# In-depth plasma *N*-glycoproteome profiling using narrow-window data-independent acquisition on the Orbitrap Astral mass spectrometer

Shelley Jager[1], Martin Zeller[2], Anna Pashkova[2], Douwe Schulte [1], Eugen Damoc[2], Karli R. Reiding [1], Alexander A. Makarov [1,2] & Albert J. R. Heck [1] ✉

Recently, a conceptually new mass analyzer was introduced by pairing a quadrupole Orbitrap mass spectrometer with an asymmetric track lossless (Astral™) analyzer. This system provides >200 Hz MS/MS scanning speed, high resolving power, sensitivity, and mass accuracy. Due to its speed, the instrument allows for a narrow-window data-independent acquisition (nDIA) strategy, representing a new technical milestone in peptide-centric proteomics. However, this new system may also be applied to other complex and clinically important proteomes, such as the human plasma *N*-glycoproteome. Here, we evaluate the Orbitrap Astral mass spectrometer for the in-depth analysis of the plasma *N*-glycoproteome and pioneer a dedicated nDIA workflow, termed "nGlycoDIA", on glycopeptide enriched and crude plasma. This strategy leads to the cumulative identification of over 3000 unique glycoPSMs derived from 181 glycoproteins in just 40 minutes and covers a dynamic range of 7 orders of magnitude for a glycopeptide enriched plasma sample. Notably, we detect several glycosylated cytokines that have reported plasma concentrations in the ng/L range. Furthermore, shortening the gradient to 10 min still allows for the detection of almost 1850 (95% CI [1840-1860]) unique glycoPSMs, indicating that high-throughput in-depth clinical plasma glycoproteomics may be within reach.

Protein glycosylation is by far the most frequent and abundant co-/post-translational modification (PTM) of plasma proteins[1,2]. Compared to other PTMs, protein glycosylation is incredibly heterogenous, with many different possible glycan compositions and even more structural variations. Moreover, protein glycosylation is highly dynamic; glycosylation affects both biological properties of the protein, e.g., receptor interactions, immune response, and protein localization, as well as its physical properties, e.g., solubility, structure, and stability[3–6]. Glycan compositions and structures can change drastically upon external factors, such as diet, disease, and aging[2–4,7]. Recent advances in untargeted *N*-glycoproteomics, primarily performed using LC-MS/MS, have led to the identification of many putative glycan biomarkers from plasma and/or serum[8–13]. Although mass spectrometry-based glycoproteomics has been advancing at an incredible pace over the last decade, it is still facing major challenges compared to standard proteomics. Among these are signal dilution caused by glycan

[1]Biomolecular Mass Spectrometry and Proteomics, Bijvoet Centre for Biomolecular Research and Utrecht Institute for Pharmaceutical Sciences, Utrecht University, Utrecht, Netherlands. [2]Thermo Fisher Scientific (Bremen) GmbH, Bremen, Germany. ✉e-mail: a.j.r.heck@uu.nl

microheterogeneity, relatively poorer ionization efficiency of glycopeptides and the need for more complicated fragmentation methods to achieve glycan and peptide sequence information (e.g., stepping-HCD, EThcD). Plasma glycoproteomics is further complicated by the unfavorable high dynamic range of proteins occurring in plasma[14–17]. So far, to obtain a deep coverage of the glycoproteome, plasma glycoproteomics has generally been performed by analyzing enriched glycopeptides with LC-MS/MS using long LC gradients and data dependent acquisition (DDA).

In recent years, however, standard proteomics has been moving away from DDA and long LC gradients, shifting towards the higher throughput enabled by data independent acquisition (DIA). In DIA, instead of a single precursor, a broader mass window is chosen, leading to the selection of multiple precursors at once that are fragmented simultaneously. The resulting fragmentation spectra are then matched to either experimentally derived or theoretically generated spectral libraries[18]. While the use of DIA is gaining prominence, particularly for high-throughput, quantitative proteomics, it is not yet prevalent for discovery-based research and glycoproteomics[19]. For instance, frequently used software algorithms adopted for DIA data analysis, e.g., DIA-NN[20] and Maxquant[21], have not yet been fully adopted and/or validated for glycopeptide library generation or annotation, although MSFragger[22] recently added a library-based DIA-glycoproteomics workflow.

There are some recent developments in the area of glycoteomics. Notably, glyco-DIA has been developed for *O*-glycoproteomics as a platform that allows library generation based on DDA data, with subsequent analysis and quantification of DIA data[23]. For *N*-glycoproteomics, GproDIA has been reported, which relies on pGlyco for library generation, subsequent DIA analysis and quantification[24]. Other developments in data analysis of DIA-derived *N*-glycopeptides include new algorithms for library generation (DIALib)[25] or the practice of reformatting DIA into DDA type files, e.g., via DIA-Umpire, whereafter the data can be analyzed using a standard DDA data processing workflow[26–28]. Another recently reported strategy is to use DIA to screen for oxonium ion features, followed by DDA for further identification of the glycopeptides[29]. Finally, Pradita et al. recently reported an untargeted DIA workflow on a purified protein, using Byonic for DIA-library generation and subsequent quantification in Skyline[30]. The latter still used manual curation for the matches generated by Skyline.

New advances in high-speed mass analyzers, such as the recently introduced Thermo Scientific™ Orbitrap™ Astral™ mass spectrometer (MS), push the field of standard peptide-based proteomics towards even more narrow windows for DIA, with MS$^2$ isolation windows as small as 2 Th[31]. These small mass windows reduce the probability of chimeric spectra (resulting from the co-isolation of different precursors exhibiting alike *m/z*) and make the resulting data very similar to DDA, which typically uses similar precursor isolation windows. In addition, the very fast duty cycle of the Orbitrap Astral instrument makes it possible to decrease the gradient length of the LC without sacrificing much depth in proteome coverage[32]. However, these powerful methods have not yet been fully tested and optimized for plasma *N*-glycoproteomics.

In this work, we aim to develop a strategy for DIA *N*-glycoproteomics on the Orbitrap Astral MS. As a benchmark to compare we use our current HCD-based workflow for plasma glycoproteomics, including an optional enrichment of glycopeptides using cotton-HILIC[33], and data analysis using Byonic[34]. With the here developed DIA *N*-glycoproteomics workflow we aim to optimally use the speed offered by the Orbitrap Astral MS, using narrow-window DIA to allow application of conventional DDA data analysis workflows. For this purpose, glycopeptides were enriched from human pooled plasma using cotton-HILIC SPE and analyzed using different gradient lengths and collision energies. Following this experimental setup, we observed

that DIA on the Orbitrap Astral MS needed to be substantially optimized for the analysis of glycopeptides, primarily because of their generally higher masses and lower charges, resulting in higher *m/z* values, and their distinct fragmentation characteristics. We here demonstrate how our optimizations led to an unprecedented coverage of the plasma glycoproteome, even when using relatively short gradients lengths of 10–20 min, where identifications of glycoproteins span about 6 to 7 orders of magnitude in plasma concentrations. For comparison, when using crude plasma, without any enrichment or depletion, we can, using the same nGlycoDIA approach, cover glycosylation of proteins spanning 3 orders of magnitude. We foresee that such glycoproteomics directed methods would enable to measure plasma glycoproteome profiles in a 20 min timeframe per sample, using only 10 μL of plasma per donor, substantially increasing the throughput and depth of these clinically relevant analyses.

## Results
### Setting up narrow-window nGlycoDIA
To set up a glycoproteomics-directed narrow-window DIA method, i.e., nGlycoDIA, several experimental parameters needed to be considered and optimized, including precursor *m/z* range, isolation window size, maximum ion injection time, and collision energy. While a standard DDA method typically uses the full MS$^1$ range (e.g., *m/z* 380–2000[33,35–37]), this would lead to an undesirably large cycle time in DIA, especially when narrow selection windows are used. Furthermore, glycopeptides often require different fragmentation strategies for optimal peptide and glycan coverage, which can be achieved with stepping collision energy[14,15,38]. However, stepping collision energy would also slow down the instrument and increase cycle time, making it less desirable for a DIA method. Because of the reasons above, it was decided to optimize the method using a smaller MS$^1$ precursor range and a single collision energy. Since data analysis platforms for DIA-based glycoproteomics are still under development, we aimed to search the DIA data as if it was DDA data. For this, Byonic was chosen as software platform as this is presently one of the most established software suites for glycopeptides[34].

To explore and optimize the best parameters, particularly the MS$^1$ precursor range and the isolation window, a series of DDA experiments was conducted. For this, a cotton-HILIC glycopeptide enriched plasma sample was measured on the Orbitrap Astral, starting with a large MS$^1$ scan range (*m/z* 300–2000) and a 2 s cycle time. The precursor isolation window was set to 1.6 Th, which is the default value. After removing all PSMs that were only found in a single sample across our quadruplicate injections, we identified ~2000 unique glycopeptide PSMs per run (Supplementary Fig. 1A). The observed glycopeptides had a substantially higher *m/z* distribution than unmodified peptides, in line with what has been reported earlier[13,35,39], likely due to their generally larger size/mass and reduced charge (Supplementary Fig. 1B). Based on this observed glycopeptide precursor ion distribution, we chose our glycoproteomics-specific DIA method to span an *m/z* range of 955–1655 (median ~1300), which is quite distinct from, and nearly does not even overlap with, the proteomics range that is typically used in (narrow-window) DIA proteomics, i.e., *m/z* 380–980 (median ~680)[31,32]. An advantage of this change in selected precursor *m/z* range is that it effectively leads to a second degree of glycopeptide enrichment, namely in the gas-phase. Testing this alternative precursor mass range in a DDA experiment, we could not only observe a 10-fold decrease in non-glycosylated peptides (*p* < 0.0001) but a 15% increase in unique glycopeptides as well (*p* < 0.01) (Supplementary Fig. 1A). We further examined the identified glycopeptides across all replicates between the different MS$^1$ ranges and found a high degree of overlap: 87% of glycopeptides identified in the default range MS$^1$ method were also identified in the method with the glycan-specific mass range (*m/z* 955–1655) (Supplementary Fig. 1C). In conclusion, the glycan-specific mass range increases the number of identified

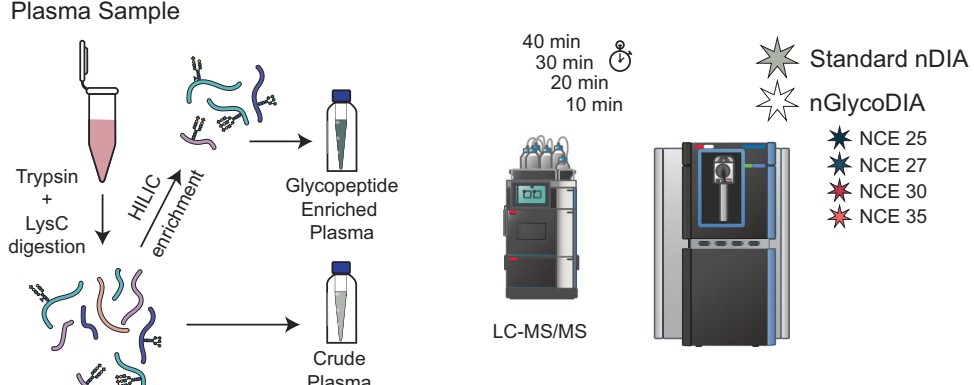

**Fig. 1 | Evaluation of data-independent acquisition (DIA) of plasma glycopeptides.** Plasma samples were proteolytically digested using trypsin and LysC. The resulting sample (crude plasma digest) was either directly subjected to LC-MS/MS analysis, or first enriched for glycopeptides by cotton-based hydrophilic interaction chromatography (HILIC). LC-MS/MS analysis was performed with different gradient lengths (10, 20, 30 and 40 min). Employed MS methods included both the standard DIA method, as typically used for non-modified peptides, as well as various optimized nGlycoDIA methods using different Normalized Collision Energies (NCEs; 25, 27, 30 and 35%).

glycopeptides, effectively depletes (co-purified) non-glycosylated peptides, and covers most of the plasma glycoproteome.

In previously published work using narrow-window DIA on the Orbitrap Astral (here further mentioned as "standard nDIA", an $m/z$ range of 380–980 was used with a 2 Th isolation window and 3 ms isolation time, leading to 300 windows and a theoretical cycle time of 900 ms. For nGlycoDIA, the isolation time was increased to 4 ms to enhance spectral quality, and to limit the cycle time, the isolation window was increased to 3 Th. With the precursor MS$^1$ range set to $m/z$ 955–1655, this led to 235 windows and a theoretical cycle time of 940 ms, which is similar to the standard nDIA method. Next, we aimed to determine the reproducibility and accuracy of Byonic with wider isolation windows (3 Th compared to 1.6 Th). Therefore, DDA experiments were performed with 3 Th isolation windows, with both mass ranges ($m/z$ 300–2000 and 955–1655). The number of unique glycoPSMs seems to increase somewhat using the 3 Th isolation window, but not significantly so ($p > 0.5$ for both MS$^1$ ranges) (Supplementary Fig. 1A). The overlap between the two isolation windows was 87% for the glycan-specific mass range ($m/z$ 955–1655), therefore we felt that Byonic could still assign the right precursor and glycopeptides with a larger isolation window (Supplementary Fig. 1D). Furthermore, the number of chimeric spectra were examined, i.e., the number of scans with 2 PSMs, which was on average 1.5% in the 1.6 Th method and 2.4% in the 3 Th method (both $m/z$ range 955–1655) (Supplementary Fig. 1E). In summary, these DDA experiments revealed that the data analysis is still reliable and reproducible when broadening the isolation window and the percentage of chimeric spectra is still low.

Next, the effect of different precursor isolation window sizes (3, 6, and 12 Th) was examined. It was decided to keep the cycle time the same in all methods, which was achieved by doubling the accumulation time along with the window size (4, 8 and 16 ms, for the 3, 6, and 12 Th windows, respectively). It was expected that an increase of isolation time would lead to higher signal and lower noise, but that the increase in window size would increase the number of chimeric spectra as well. The experiment revealed that the unique glycosylated PSMs increased from around 2200 to around 2600 per injection with the 6 and 12 Th windows (Supplementary Fig. 2A). However, the increase in accumulation time did not lead to overall higher Byonic scores, suggesting a minimal effect on spectral data quality (Supplementary Fig. 2B). At the same time, the number of chimeric spectra increased from 2.6% in the 3 Th method (comparable to the DDA method), to 5.3% in the 6 Th and 8.9% in the 12 Th method (Supplementary Fig. 2C). Because we relied on DDA based software in the analysis, we decided to prioritize

confidence over the number of identifications, and thus decided to move forward using the 3 Th isolation window. Finally, the AGC target was increased to 800% to fully use the 4 ms injection time.

## nGlycoDIA enables deep coverage of low abundant plasma proteins

Next, using the experimental parameters described above, a set of DIA methods were designed for the Orbitrap Astral MS exploring four different NCEs, namely 25, 27, 30, and 35%. Because DIA often results in identification of low abundant proteins and increased sensitivity[40,41], we wanted to explore how many glycoproteins could already be observed without glycopeptide enrichment. Therefore, we analyzed in parallel both plasma samples from which the glycopeptides had been enriched by cotton-HILIC, as well as what we term here "crude" plasma samples (Fig. 1). Additionally, as shown for standard proteomics with the introduction of 40 up to 180 samples per day (SPD) methods[31], we hypothesized that the increased speed of the Orbitrap Astral MS would allow us to shorten the LC-MS runs, without a significant loss in identifications. We therefore evaluated four different LC gradients, going down from 40, 30, 20, to even 10 min.

First, the 40-min gradient results were examined because we expected to identify the highest number of glycopeptides using this longest gradient. Higher fragmentation energy led to the identification of more unique glycopeptide PSM (glycoPSMs), their number steadily increasing across NCEs 25–35% (Fig. 2A). This increase is attributed to both an increase in identified proteins and glycosites (up to NCE 30%), as well as an expansion in detected glycan microheterogeneity per site (Fig. 2B, C). These NCE-dependent features were observed in measurements for both the crude and enriched plasma samples. Overall, the glycopeptide enrichment led to a substantial increase in unique glycopeptide identifications, compared to crude plasma. In the standard nDIA method, a significantly lower number of glycopeptides was identified, most detections being non-glycosylated peptides instead. This is also represented in the number of glycoproteins and glycosylation sites. Interestingly, the increase in the number of co-enriched non-glycosylated peptides lead to an increase in non-glycosylated PSMs compared to the crude plasma, which is also represented in the number of identified proteins in each sample (252 in the enriched plasma compared to 184 in the crude plasma). Injection replicates were highly reproducible, with approximately 80% of identifications found in at least two replicates (Supplementary Fig. 3A, B). Furthermore, most glycopeptides that were identified in two injection replicates were found in all four of the replicates (Supplementary Fig. 3A). When

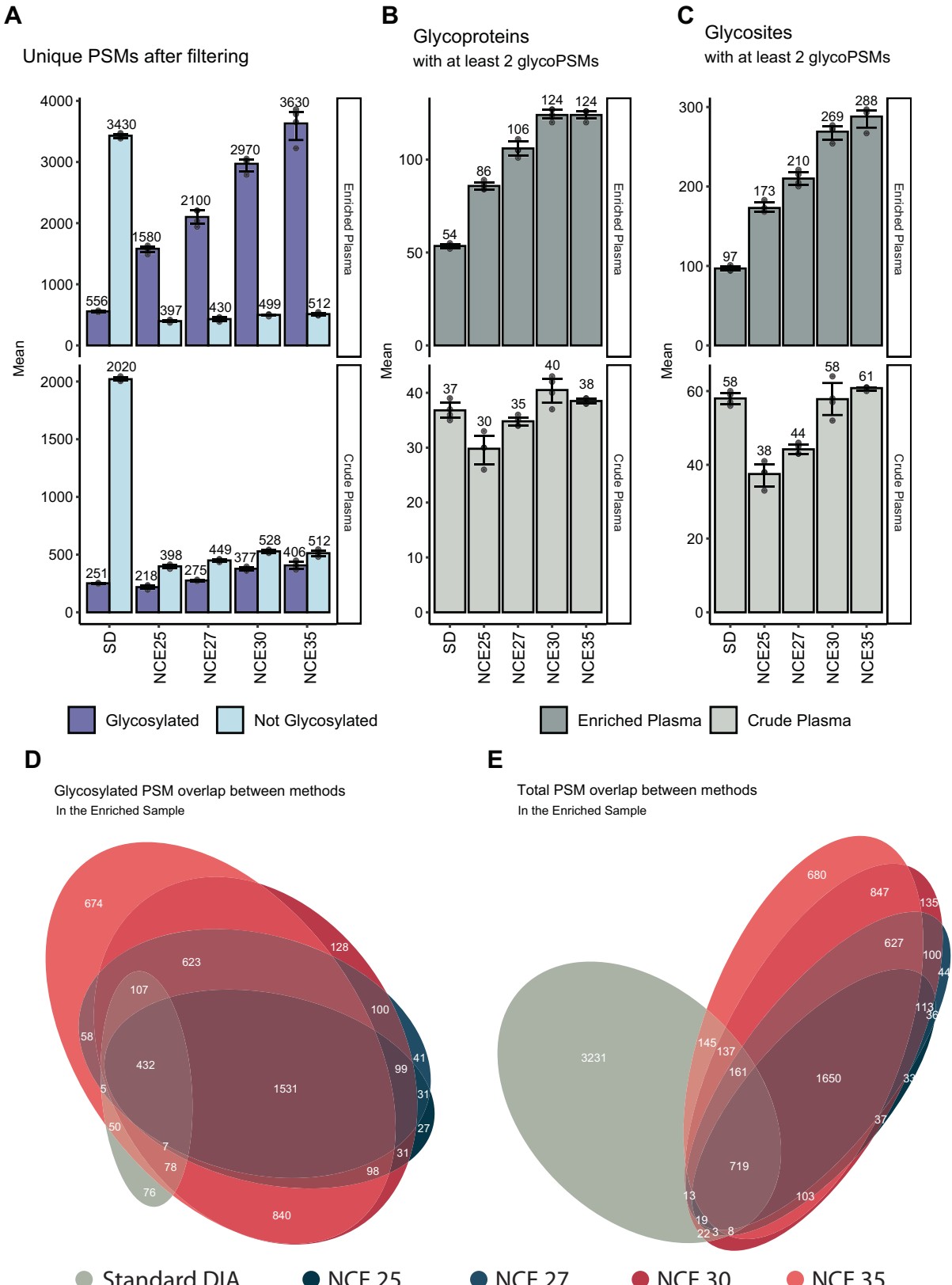

comparing between NCEs, the overlap between the smaller datasets (acquired with lower NCE) was similar to the overlap between injection replicates, indicating that the increase in annotatable spectra was an increment and not a bias (Supplementary Fig. 3B).

Next, the identifications extracted from the analysis of the crude plasma were compared to those identified in the glycopeptide-enriched plasma. Here we see that approximately 85% of the glyco-peptide identifications in crude plasma were also found after enrichment. Overall, a high overlap (only 1–4% unique glycoPSMs per method, excluding NCE 35%) of identified PSMs between the different nGlycoDIA methods was observed for both glycosylated and unmodified peptides, indicating high method robustness (Fig. 3D, E). Almost

**Fig. 2 | nGlycoDIA glycoproteomics data obtained for crude and glycopeptide-enriched plasma samples.** The top three panels depict data for glycopeptide-enriched plasma, whereas the middle three panels depict data for crude plasma. **A** The number of unique PSMs for glycopeptides (dark blue) and non-modified peptides (light blue). **B** The number of proteins identified. **C** The number of glycosites identified. The consecutive bars reflect the results obtained with the different DIA methods and NCE energies, as annotated. Four technical replicates were performed for each method. The bars illustrate the mean values, while the error bar represent the 95% confidence interval. The x-axis annotation on (**A**–**C**) refer to the MS/MS methods, where SD is standard DIA, and NCE25, NCE27, NCE30, and NCE35 refer to nGlycoDIA with the respective NCE percentages. **D** A Venn-diagram of the PSM overlap between methods for the detected glycosylated PSMs. **E** A Venn-diagram of the overlap of PSMs between methods considering all PSMs both glycopeptides and non-modified peptides. The colors correspond to the used MS-methods: standard DIA (gray), and nGlycoDIA with NCE 25 (dark blue), NCE 27 (blue), NCE 30 (red), NCE 35 (salmon). All data (**A**–**E**) was filtered for 1% FDR and, to be included, each unique PSM had to be identified in at least two out of four injection replicates for a single condition. Source data are provided as a Source Data file.

all identified glycosylated PSMs found in the standard DIA method (90%) were detected in at least one of the nGlycoDIA methods as well. This illustrates that the data lost by shifting the precursor $m/z$ range is minimal, and that this rather results in a beneficial additional degree of enrichment in the gas phase, as was demonstrated in the aforementioned comparison with DDA.

While the number of uniquely identified glycoPSMs kept increasing with higher collision energies, we observed that the quality of the MS$^2$ spectra decreased from 25% NCE to 35% NCE. This especially became evident once examining the glycan specific ion coverage over the entire dataset (Supplementary Fig. 3C). Both B- (oxonium) and Y-ion coverage decreased consistently over different precursor charge states. Figure 3 serves as an example, demonstrating that by using lower collision energy, larger glycan fragments, as well as abundant Y-ions were preserved, which increased the confidence in both glycan and peptide assignment. These Y-ions are crucial when modifications of the peptide backbone might overlap in mass with different glyco-forms. For instance, the mass difference between a hexose and a fucose is identical to that of a peptide with and without oxidation, i.e., 15.99 Da, the mass of an oxygen. Additionally, while Y-ions are precursor-specific, oxonium ions are not. As such, Y-ions are critical in determining glycopeptide identity in the case of chimeric spectra. As the amount and quality of Y-ions in our data showed a clear drop-off after 30% NCE, we opted for 30% NCE as the most ideal collision energy for nGlycoDIA.

Next, we used the blood protein concentrations reported in the Human Protein Atlas (which we will refer to as the Blood Atlas[42,43]) to explore the depth of proteome coverage we could reach using nGlycoDIA. In the crude plasma, primarily highly abundant glycoproteins (concentration > 1 mg/mL) were identified (Fig. 4A, B). When comparing standard DIA to our optimized nGlycoDIA method, it was clear that in the latter not only more glycoproteins in the mg/L range were identified (51 with nGlycoDIA compared to 42 with standard DIA), but even lower-abundant glycoproteins such as Factor VIII and the epidermal growth factor receptor. Additionally, the glycan microheterogeneity that could be identified was overall higher when using nGlycoDIA (Fig. 4D). The glycans characteristics observed in the standard DIA method were also quite distinct and shifted towards lower mass glycans when compared to the nGlycoDIA methods. For example, in standard DIA we observed more N4H5S1 than N4H5S2, relatively more paucimannose glycans, and almost no tri- and tetra-antennary glycans (Supplementary Fig. 4). This was not surprising, because the DDA experiments had already demonstrated that most glycopeptides fall outside of the precursor range used in the standard DIA method. It should be noted that the method presented here focused on plasma *N*-glycoproteomics, where large complex-type glycans are highly abundant. When analyzing different types of samples where smaller glycans are more prevalent, such as paucimannose glycans, the MS$^1$ scan range might require additional optimization. There was no clear difference in glycan distribution between the different NCE conditions used, and the distribution of different glycans was very similar to other plasma glycomics and glycoproteomics studies, with glycopeptides bearing mostly sialylated complex type glycans with two antennae, followed by complex tri and tetra-antennary glycans[6,13,35,44].

Following HILIC enrichment, substantially more glycoproteins were identified and consequentially a larger dynamic range was covered (spanning 7 orders of magnitude). Compared to the analysis of crude plasma with nGlycoDIA, the number of glycoproteins detected in the enriched sample in the mg/mL range was doubled, in the μg/mL range the number of identifications increased by over 4-fold and we were even able to identify 17 glycoproteins in the ng/mL range (Fig. 4C). Additionally, 12 glycoproteins were identified that only have a reported quantification in the Blood Atlas by immunofluorescence and not by mass spectrometry. Moreover, 25 glycoproteins were detected that have not been quantified in the Blood Atlas at all, amongst them being IGHD, ITIH5, and intriguingly even several cytokines which we will discuss in more detail in the following section. As these proteins are currently not quantified in the human blood atlas, they cannot be shown in the S-curves in Fig. 4. Next to increased detection of low-abundant proteins and glycosylation sites, for the glyco-enriched plasma we were able to detect more glycan structures per site compared to its crude counterpart. In the most extreme case, we observed more than 80 different glycans on a single site, namely N639 of sero-transferrin, highlighting the visibility of glycosylation heterogeneity (Fig. 4D). Overall, the data demonstrate that enrichment for glyco-peptides is still highly beneficial for the analysis of low abundant glycoproteins (concentration <1 mg/mL) in a complex matrix such as plasma.

## Benchmarking plasma glycoproteomics *versus* proteomics, detection of glycosylated low abundant cytokines

In the present study we identified 181 glycoproteins and 436 unique glycosites with the optimized nGlycoDIA approach, taking as a cut-off their detection in at least two out of four individual experiments (acquired at different NCE). We were intrigued by the detection of several glycosylated cytokines and growth factors in our study, whereby we here focus on and discuss only the ones that we detected in at least two experiments. These include the interleukins IL-12A, IL-12B and IL-22, the colony stimulating factors CSF1 and CSF2, and the vasoactive intestinal peptide (VIP). All these cytokines are known to play an eminent role in inflammation and diseases such as cancer[45–48], but experimental evidence for their glycosylation in human plasma has as far as we know never been reported. The here provided insight into their glycosylation can provide a valuable background for elucidating the role of these PTMs in the disease mechanisms in which these cytokines play a role. Below we will describe in more detail the glycosylation features that we identified on these cytokines, which we nearly exclusively identified using the nGlycoDIA method. Therein, we aimed to validate the glycan composition and structural elements by manually assessing the spectra (Fig. 5A). Furthermore, to demonstrate the relevance of these glycans, we modeled the most frequently identified glycan composition on structural models of the proteins, when available.

First, we focused on the interleukins. Interleukin (IL) 12 is a potent immunoregulatory cytokine crucial for the generation of cell-mediated

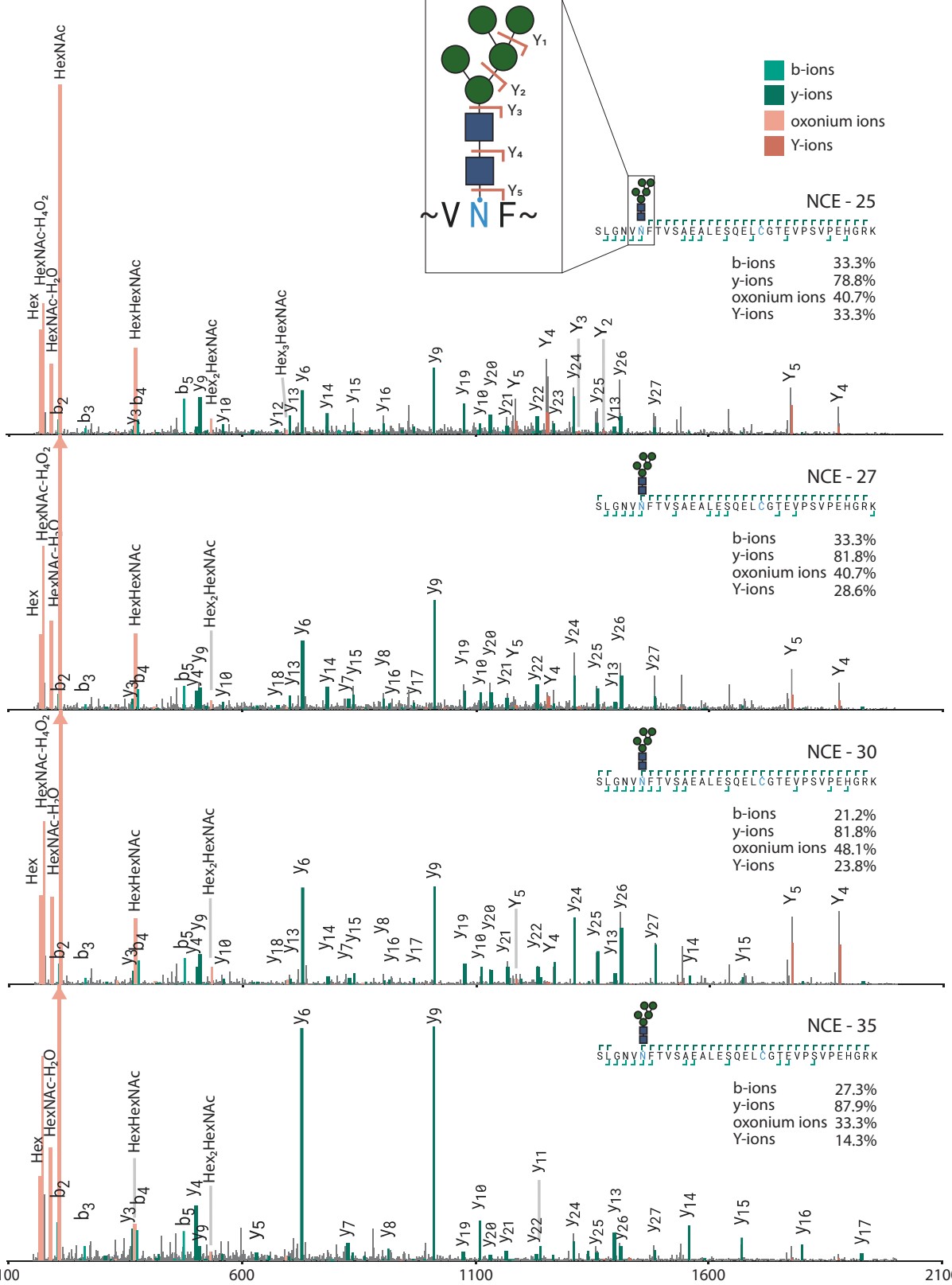

**Fig. 3 | Effect of collision energy on glycopeptide fragmentation spectra.** The highest scoring PSM for each specific peptide with glycan was selected per condition to ensure a fair comparison. In the figure, b- and y-ions are represented in light and dark green, respectively, while B- (oxonium) and Y-ions are depicted in light and dark orange, respectively. In the top right corners, the coverage of the peptide backbone is shown, with percentages of total possible positions covered by the respective ion. Annotations were performed using Annotator (v 0.2.2)[59].

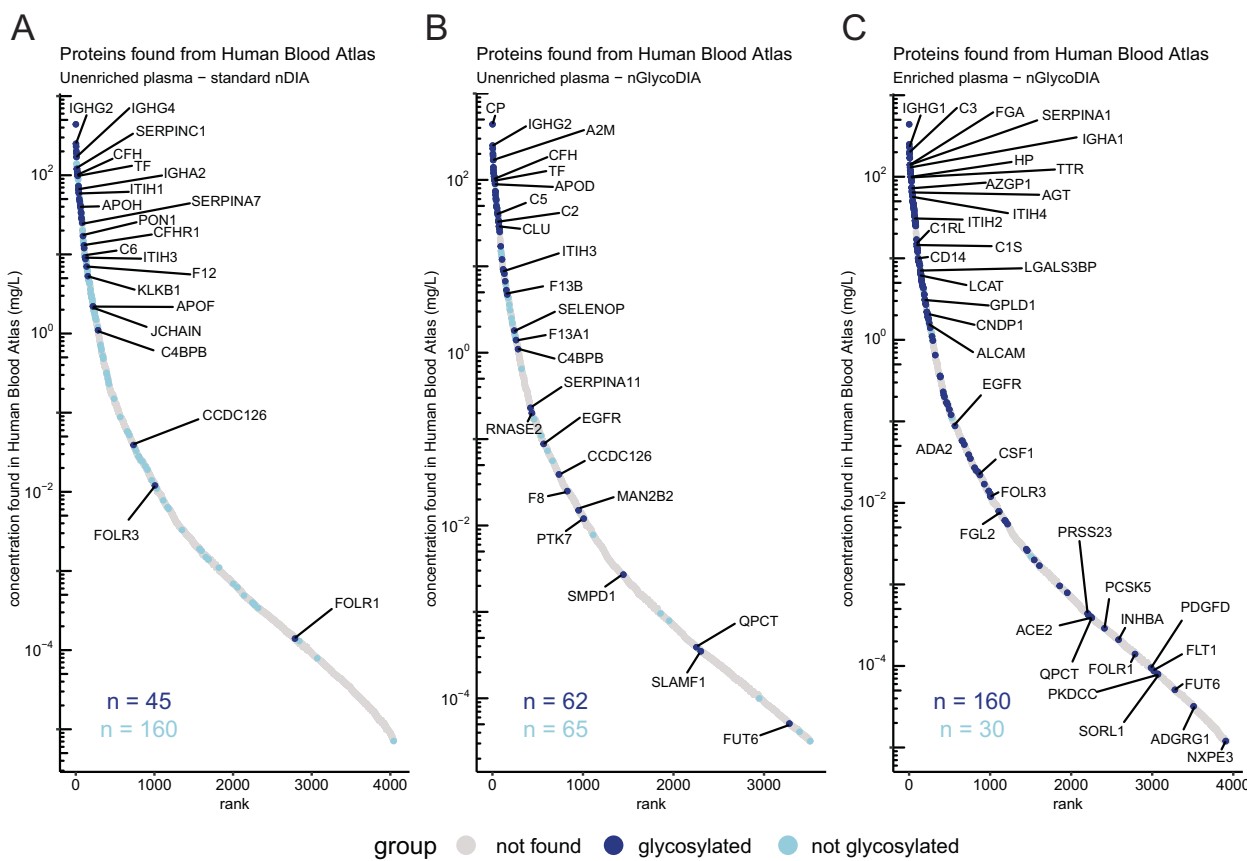

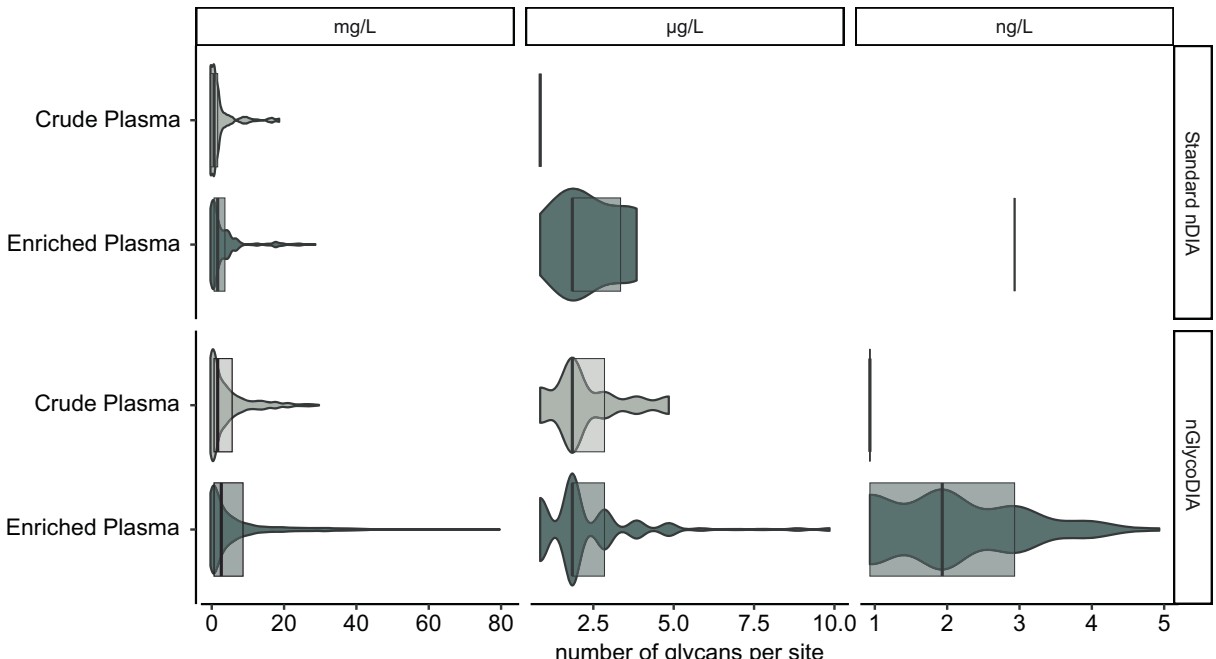

immunity to intracellular pathogens[47]. It is a hetero-dimer, consisting of the sub-units IL-12A and IL-12B, of ~25 and ~37 kDa, respectively. For IL-12A we identify five different glycan compositions of complex type glycans on N217, a site that is however not annotated as a putative glycosite in UniProt. The detected glycans are N4H5S1, N4H5F1S1, N5H6S2, N5H6F1S2, and N5H6S3. Most of them seem to have one sialic acid less than the number of antennae. We were unable to establish whether the additional arm was branched or elongated. Manual analysis of the Y-ions in the different nGlycoDIA spectra did confirm that the fucose was most likely located on the core HexNAc. This N217 carries the *N*-glycan consensus motive and is located very near the C-terminus of IL-12A, (NAS), whereby Ser represents the C-terminal

**Fig. 4 | Evaluation of the depth and annotatable microheterogeneity of both high abundant and low abundant plasma glycoproteins following analysis by nGlycoDIA. A** the proteins detected in crude plasma using the DIA method optimized for standard proteomics. **B** Protein detections when analyzing crude plasma digest by nGlycoDIA. **C** Protein detections following glycopeptide enrichment and subsequent analysis with nGlycoDIA. All detected proteins were compared to their reported abundance in "normal plasma" as described for the blood proteins of the Human Protein Atlas (Blood Atlas), as measured by MS[42,62]. In the bottom-left corner, the number of proteins detected with and without identified glycosylated peptides is indicated in dark blue and light blue, respectively. The colors of the dots in (**A–C**) represent the following: gray – not identified with nGlycoDIA, light blue – non-glycosylated peptides identified with nGlycoDIA, dark blue – glycosylated peptides identified with nGlycoDIA. Several dots are annotated with the corresponding gene name, genes to annotate were chosen at random using the ggrepel package (v 0.9.5). **D** Detected glycan microheterogeneity depicted as unique glycans identified per glycosite in crude (light green) and enriched dark green) plasma,

analyzed using standard nDIA and nGlycoDIA. Glycosites with only 1 psm per run were removed for this analysis. Number of glycans per site is analyzed per individual run, and the values of the 4 injection replicates are combined for each violin. Data is grouped on the reported abundance of the proteins in the Blood Atlas, where mg/L ranges between 999 – 1 mg/L, μg/L between 999 – 1 μg/L, and ng/L below 999 ng/L. Data is displayed as a violin plot, which extends from the minimum and maximum value identified. Overlayed is a boxplot with the center line at the median value, upper bound at the 75th percentile and lower bound at the 25th percentile, without whiskers. The number of glycosites in each group was as follows: crude plasma standard nDIA—mg/L: $n = 229$, μg/L: $n = 1$; enriched plasma standard nDIA—mg/L: $n = 362$, μg/L: $n = 7$, ng/L: $n = 1$; crude plasma nGlycoDIA—mg/L: $n = 728$ μg/L: $n = 44$ ng/L: $n = 5$; and enriched plasma nGlycoDIA—mg/mL: $n = 3312$ μg/L: $n = 199$, ng/L: $n = 102$. The comparison revealed that nGlycoDIA in glycopeptide-enriched plasma led to most glycans detected per site, allowing the detection of glycoproteins across a dynamic range of $10^6$. Source data are provided as a Source Data file.

amino acid. We could trace no earlier structural or functional data on this intriguing C-terminal *N*-glycosylation.

For IL-12B we detected two distinct glycosylation sites, namely at N125 and N135. For site N125, we only detected a peptide with two amino acids, and although we detected two different glycans on it across multiple conditions, we felt this data is not sufficient to confidently assign the site. On site N135, which is annotated as putative glycosite in UniProt containing the consensus motif (NYS), we detected several hybrid type glycans (Supplementary Fig. 5A–D). These glycans contained at least one sialic acid and some were fucosylated with up to two fucose moieties as well. One of the glycoforms was consistently annotated as N4H5S2, however, after manual curation, we corrected this to N4H5F2S1 (Supplementary Fig. 5B). This is a well-known mistake that occurs frequently when the annotation software picks the wrong peak during monoisotopic peak assignment. Another interesting feature on these hybrid glycans was the direct loss of an *N*-acetylhexosamine from the precursor upon fragmentation, suggesting the presence of a bisecting GlcNAc or additional antenna.

We mapped the sites and glycan structure supported with the most glycoPSMs in our study for IL-12A and IL-12B (N4H5S1 and N3H6S1, respectively) on the reported X-ray structural models of the IL-12 heterodimer (PDB: 1F45) (Fig. 5B)[49]. While we could estimate several glycostructural aspects from the fragmentation spectra, the exact structures were assumed on basis of the literature on plasma glycosylation[6,50]. Only the site on IL-12A could be modeled on the dimer, as the site for IL-12B was not in a solvent-accessible site in its configuration within the hetero-dimer. We were able to map the glycan on IL-12B on its monomeric configuration (PDB: 1F42) and overlaying it to the hetero-dimeric conformation (Fig. 5C). Taking a closer look at the structure of the hetero-dimer and the glycans, we noticed that the core GlcNAcs from IL-12A-N217 formed hydrogen bonds with the sidechain of the Gln42 residue of IL-12A, which may aid in stabilizing its conformation upon hetero-dimerization (Fig. 5B). The glycan core of IL-12B-N135 also formed intermolecular hydrogen bonds with the Thr264 sidechain of IL-12B and the second GlcNAc formed a hydrogen bond to the amide of the backbone at Glu89 of IL-12A, which is in a flexible region, possibly aiding in stabilization. For the X-ray studies by Yoon et al.[49], the protein was obtained by recombinant production in CHO cells and did contain glycans. In the structure of IL-12B another *N*-glycan was reported on IL-12B at site N222 (N200 by the author's numbering), which we did not detect in our study.

On IL-22, we experimentally detected two glycosylation sites, namely at N54 and N97. On both sites we only identified one glycan: N4H5S1 on site N54 (NRT), N5H6S3 on site N97 (NFT). Both these sites are annotated in NextProt as putative *N*-glycosylation sites. IL-22 is a 20 kDa α-helical cytokine that binds to a heterodimeric cell surface receptor composed of the IL-10R2 and IL-22R1 subunits. We mapped

these two observed glycans on the structure of IL-22 when bound to the IL-22R1 receptor (PDB 3DLQ) (Supplementary Fig. 6A)[51]. The glycan at N97 points away from the complex, while the glycan at N54 is pointing towards the receptor, especially with the glycan antenna. Previously, it has been reported that glycans or IL-22 (recombinantly produced in insect cells) did not interfere with the binding of IL-22 to its receptor and that the presence of absence of the glycans does not have a large impact on the conformation[52]. The glycans on these recombinant proteins were different from those detected here in human serum; the insect cell glycans were shorter and did not have antennae, while the here detected glycan is di-antennary and has a single negative charge on the sialic acid residue. Still, also in our model the glycan likely does not influence the binding to the receptor as it is not in proximity.

The macrophage colony-stimulating factor 1 (CSF1) promotes the release of pro-inflammatory chemokines, and thus plays an important role in innate immunity and inflammatory processes. This ~60 kDa protein does carry more than a dozen *O*-glycosylation sites (not studied here) but has according to NextProt two putative *N*-glycosylation sites (N154 and N172, annotated by consensus sequence). Here, we did not find evidence for these two sites, albeit we confidently identified *N*-glycosylation at N381 (NHT), a site not annotated in NextProt. We found the site to be occupied with a N4H5S2 glycan, a di-antennary complex type glycan. There is an *O*-glycosylation site annotated in NextProt at the adjacent T383 site, located on the same tryptic peptide as N381, however the Y-ions series we identified supports that the annotation the *N*-glycan site at N381 is likely.

CSF2, also known as granulocyte-macrophage colony-stimulating factor, is a ~15 kDa protein and has in NextProt also two putative *N*-glycosylation sites (N44 and N54), of which we here identified N54 to be occupied by various complex glycans, almost always fully sialylated and sometimes harboring also core fucosylation. For this cytokine at site N54 we therefore have multiple pieces of evidence that it can be detected in plasma (reported concentration in the Blood Atlas of $2.4 \times 10^4$ pg/L) and that it is glycosylated.

We also detected the vasoactive intestinal peptide (VIP), which has no annotated glycosites in NextProt and a reported concentration of $1.2 \times 10^5$ pg/L in the Blood Atlas. On VIP we identified two *N*-glycosylation sites: N68 (NDT) and N133 (NYT). The former is located on the propeptide and the latter is located on the vasoactive intestinal peptide. Here we detected glycopeptides whereby N68 was occupied by an N4H5S2, while N133 was found to be occupied with N5H5S2. Recently, a cryoEM structure was reported revealing the interaction of a VIP peptide (125–152) coupled to its G protein-coupled pituitary adenylate cyclase-activating polypeptide (PACAP) type 1 receptor (PAC1R). The authors compared the binding between PAC1R and VIP to the binding of PAC1R to PACAP, the latter having a 1000-fold higher affinity while

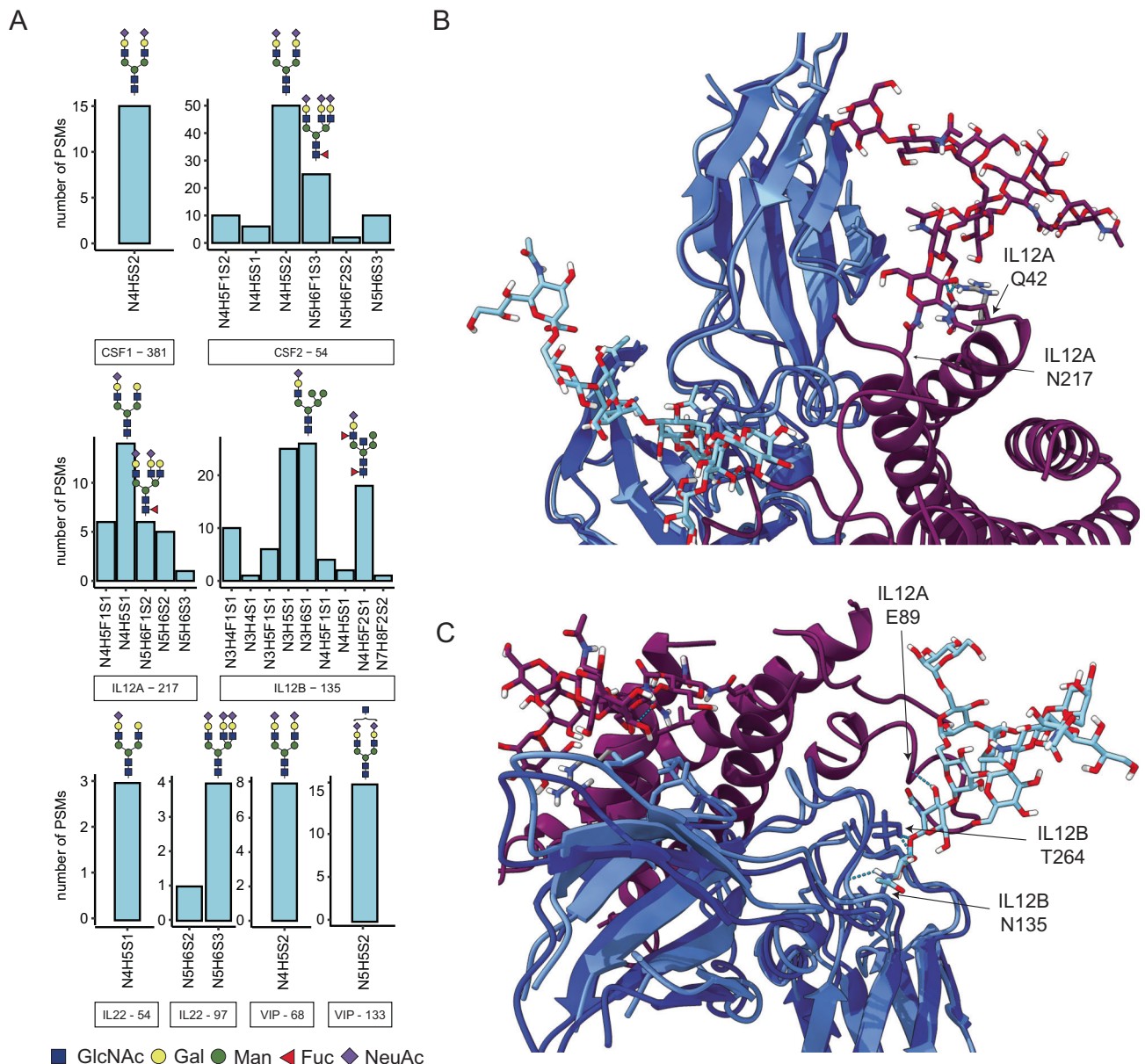

**Fig. 5 | GlycoPSMs detected on low abundant cytokines in MS-based plasma proteomics. A** The number of PSMs depicts the total number of identified glyco-PSMs across all four replicates and four NCE conditions in glycopeptide-enriched plasma. The protein name is indicated below the bar graph and the most frequently identified glycan structures are shown. **B, C** X-ray structural model of IL-12, with the here most observed glycans mapped onto these structures, where IL-12A and its glycan (N4H5S1) are purple, and IL-12B and its glycan (N3H6S1) are blue, with in light blue the protein structure as reported in the heterodimeric conformation (PDB 1F45), and in dark blue the overlayed monomeric structure (PDB 1F42). Light blue dashed lines represent possible hydrogen bonds between the glycans and proteins. The arrows indicate the glycosite and the interaction partners. Note that our glycopeptide analysis yields compositional and not structural information on the glycans. Source data are provided as a Source Data file.

sharing nearly 70% sequence identity[53,54]. Interestingly, one of the sequence differences between VIP and PACAP is in the *N*-glycan consensus motif, although the binding between PAC1R and VIP was only investigated using a non-glycosylated VIP peptide. The VIP peptide agonist was shown to bind in the relatively large *N*-terminal extracellular domain (ECD), facilitating engagement of the peptide N-terminal activation domain with the extracellular loops (ECLs) and transmembrane helices (TMs) of the receptor core. We modeled the here observed glycan on this structure (PDB 8E3Z) (Supplementary Fig. 6B). As the peptide is fully situated within the ECD domain, with the glycan sticking out, the here reported glycan moiety very likely affects this binding.

Although they are normally extremely low abundant, we detected several more cytokines exclusively by our nGlycoDIA approach, and only in the glycopeptide enriched plasma, albeit with less evidence. These included IL4 (N47 and N114), IL34 (N77), IL24 (N127), FMS-like tyrosine kinase 3 ligand (FLT3LG) and tumor necrosis factor superfamily member 11 (TNFSF11). The fact that we detect all these cytokines was rather remarkable as most of them have concentrations in the order or $10^4$–$10^5$ pg/L in plasma of healthy donors (as analyzed here), whereas the concentration of, for instance, the immunoglobulins and some of the serpins are $10^{10}$–$10^{11}$ pg/L in plasma. This demonstrates an apparent dynamic range of $10^6$ in glycoprotein abundance using our nGlycoDIA approach on enriched plasma.

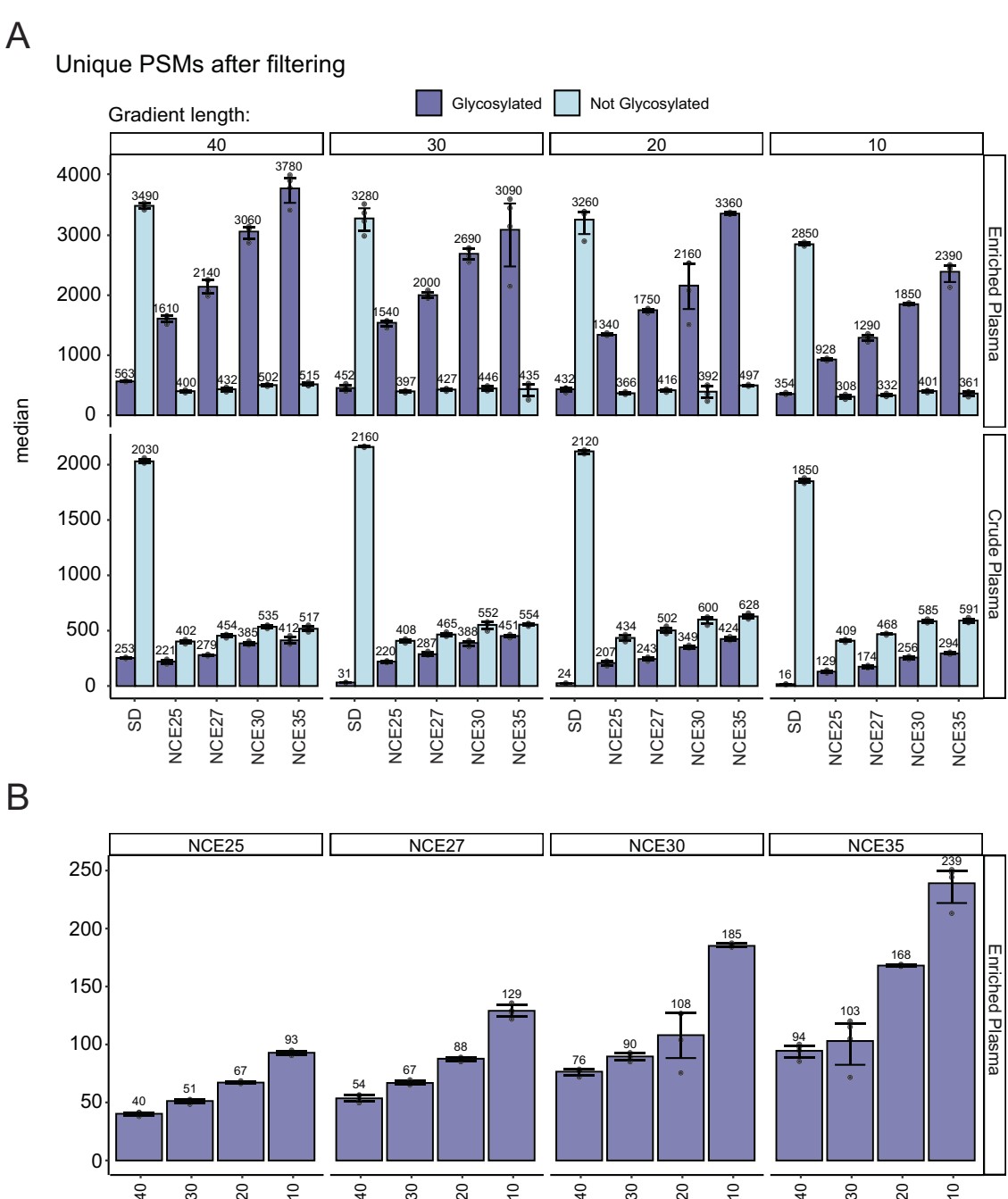

**Fig. 6 | Evaluating the time-efficiency of the DIA glycoproteomics methods.**
**A** Unique glycosylated (dark blue) and not glycosylated (light blue) PSM count identified per method across four technical replicates in enriched plasma (top-panel) or crude plasma (bottom-panel). The bars illustrate the mean values, while the error bars represent the 95% confidence intervals. The x-axis annotations refer to the MS/MS methods, where SD is standard DIA, and NCE25, NCE27, NCE30, and NCE35 refer to nGlycoDIA with the respective NCE percentages. **B** Unique

glycosylated PSMs per minute in enriched plasma using different MS$^2$ NCEs. The PSMs per minute is calculated as the PSM count divided by gradient length. The bars illustrate the mean values, while the error bar represent the 95% confidence intervals. All data is filtered for 1% FDR and each unique PSM was identified in at least two out of four injection replicates for a single condition. Source data are provided as a Source Data file.

## Towards high-throughput and high-sensitivity plasma glycoproteomics

Although 40-min gradients are already relatively short for glycoproteomics, to investigate larger clinical cohorts of plasma samples for glycoprotein biomarkers any reduction in analysis time would be beneficial. With the optimized nGlycoDIA method in hand, an effort was made to decrease the time of each run. We tested the optimized nGlycoDIA method with different gradient lengths (40 min, 30 min,

20 min and 10 min). In fairness, note that these numbers represent the LC gradient length, and not the total time of the LC method as the latter includes an additional 6-min column wash.

In the glycopeptide enriched plasma, we could detect a decrease in the number of unique glycopeptide PSMs with shorter gradient times. Still, just about 10% of all unique PSMs were lost when the gradient time was reduced from 40 to 20 min, while ~40% was lost when decreasing the gradient time to 10 min (Fig. 6A). The number of

annotated PSMs without glycan decreased at maximum 25% from 40 to 10 min when using nGlycoDIA in enriched plasma. For the crude plasma, our data indicated that extending the gradient past 20 min did not lead to an increase of unique glycoPSMs. The number of proteins and glycosites revealed similar trends. Another way to examine the time-efficiency is to look at the unique identifications per minute (Fig. 6B). Consistently for all methods, the glycosylated identifications per minute decreased with gradient duration, the 10-min gradient being still the most efficient. As far as we know these represent some of the most data-rich plasma glycoproteomics data with such short (10 min) gradients.

The overlap between cumulative unique PSMs identified in the different runs (sample type, gradient length, and NCE) were examined via a clustered heatmap (Supplementary Fig. 7). When clustering all the data, the first separation was based on enriched vs. non-enriched plasma, making it evident that the identified glycopeptides from enriched plasma overlapped highly (-80–90%) between different methods (NCEs and gradients), which was also the case for the crude plasma (-80%). Furthermore, the crude plasma has a similarly high overlap in identified glycopeptides with the enriched plasma (-80–90%), while the other way around the overlap was low (<20%), demonstrating and effective enrichment and only limited loss of glycopeptides. Subsequent grouping was primarily based on the number of identifications in each dataset, further demonstrating the reproducibility of the nGlycoDIA approach.

Lastly, we compared the dynamic range between the 10 min and 40 min gradients. Here we noticed that in the 10 min approach, compared to 40 min, we detected fewer low-concentration glycoproteins (Supplementary Fig. 8A), i.e., 32% fewer proteins in the µg/L range and almost 50% fewer proteins in the ng/L range. Additionally, there was a decrease in annotatable microheterogeneity in the µg/L range, albeit not in the other concentration ranges (Supplementary Fig. 8B).

### Compatibility with alternative data analysis software

It is well known, especially in glycoproteomics, that search engine results can vary greatly between different developers and settings[55]. Therefore, apart from using the search engine Byonic, which is quite well-established for glycoproteomics, we also explored data analysis using the alternative FragPipe suite. While the available version of FragPipe was not able to analyze the data in the same way as Byonic, i.e., by identifying the MS$^1$ precursor mass and treating the DIA-data as DDA-data, we were able to make use of two alternative strategies: 1) using DIA-Umpire and 2) using an experimental library-based DIA search. A priori, it can be expected that either strategy will yield fewer unique glycoPSMs. DIA-Umpire transforms DIA data into pseudotandem MS spectra by correlating the chromatographic features within the MS$^1$ (precursors) and MS$^2$ (fragments)[28], but this requires highly correlated features across the entire peak over multiple scans. Because the chromatographic peaks of glycopeptides often split due to glycan isomerism, which also affects their fragmentation (especially the intensities of oxonium ions)[56], we expect that the correlation between precursor and fragment features will be limited and the number of PSMs will decrease. At the same time, library-based DIA cannot conceptually outperform its DDA library, meaning that the possible number of identifications is limited as well. To nevertheless estimate the effect of the different search strategies, we performed parallel searches on our data (glycopeptide enriched plasma, 40-min gradient, 30% NCE, unless stated otherwise).

Using DIA-Umpire followed by MSFragger-Glyco, roughly 750 unique glycoPSMs were identified per run, culminating in a combined total of 904 unique glycoPSMs after filtering for a minimal occurrence of 2 out of 4 replicates (Supplementary Fig. 9A). This is notable lower than the search with Byonic (around 4500 glycopeptides cumulatively identified over four injection replicates). Of the glycopeptides identified in the MSFragger search, 829 (92%) were also identified in the

Byonic search (Supplementary Fig. 9B). Next to this, the data was searched using the default glyco-N-DIA workflow in FragPipe, which required a spectral library. We generated this library using the four previously described DDA data files (gradient of 40 min and NCE of 30%). However, as only modification with a UniMod identifier could be added to the library, this led to the loss of approximately 1700 identifications and a final library consisting of 2115 glycoPSMs. Searching the nGlycoDIA data with this library resulted in around 1600 unique glycoPSMs per run, and a total of 1737 unique glycoPSMs after filtering for a minimal occurrence of 2 out of 4 replicates (Supplementary Fig. 9A). Of these, 1061 glycoPSMs were also found by our Byonic workflow (61%) (Supplementary Fig. 9C), while 314 glycoPSMs were both identified in the DDA library and the Byonic search but were not found by spectral matching. Overall, all three software approaches (Byonic, DIA-Umpire, and N-glyco DIA in FragPipe) proved usable for data analysis of the nGlycoDIA data, although from all tested approaches Byonic provided by far the most IDs.

An interesting feature of the glyco-N-DIA workflow in FragPipe is that it facilitates the quantification in DIA-NN using the generated library. Therefore, using this method, we investigated the reproducibility in glycopeptide quantification on quadruplicate analysis of both the 10- and 40-min gradients. These analyses revealed a high correlation between replicates (Pearson R > 0.9) and between gradients (Pearson R > 0.8) (Supplementary Fig. 9D). The median CV was 2.2% and 2.3% in the 10- and 40-min methods, respectively (Supplementary Fig. 9E). Although quantification appeared robust, it should be noted that at the time of this writing the DIA workflow in MSFragger has not yet been validated by peer review, and quantitation here is only based on the most abundant glycopeptides.

## Discussion

In this study, we evaluated the novel Orbitrap Astral MS instrument for plasma glycoproteomics, demonstrating the potential of narrow-window data-independent acquisition (nDIA) for analyzing glycopeptides (nGlycoDIA). With this method, using crude plasma, we were able to annotate around 400 unique glycosylated PSMs per run (95% CI [364–405], 30 min using 30% NCE), covering 60 glycoproteins and 186 glycosites. Remarkably, without enrichment, the 20-min method was already sufficient to maximize detection of glycopeptides in this sample. However, following a simple and cost-effective cotton-HILIC based glycopeptide enrichment, nGlycoDIA resulted in the identification of around 3000 (95% CI [2940–3130]) unique glycosylated PSMs per injection replicate using 40-min LC gradients, covering in total 181 glycoproteins and 436 glycosites. These 181 proteins spanned a dynamic range across 7 orders of magnitude, significantly surpassing recent studies which reported the identification of roughly 200 unique glycopeptides using longer gradients (60–80 min)[13,44]. However, since there are many differences between the studies, it is not certain which factors contribute to the increase, be it the sample, the enrichment protocol, the chromatography conditions, the increased speed of the Orbitrap Astral instrument, and/or the nGlycoDIA method. Because of this, it is most fair to compare the nGlycoDIA results to the DDA results using the same machine, column, gradient, and collision energy. In making this comparison, we find that the number of unique glycopeptides found with nGlycoDIA is increased with 50% compared to the DDA analysis.

This is as far as we are aware one of the early applications of DIA for plasma glycoproteomics and provides a substantial advance over previous methods used in other and our laboratories. This advance can be largely contributed to the higher speed of the Orbitrap Astral analyzer, but also the use of the for glycopeptides optimized narrow-window DIA, allowing the data to be analyzed by a DDA-like computational processing workflow. DIA is for glycoproteomics much more challenging than for 'standard' proteomics, due to many factors: amongst them are the fragmentation efficiency of glycopeptides and

the high abundance of oxonium ions that are not specific to unique glycopeptides. As this is the first nGlycoDIA method for plasma glycoproteomics, and one of the first benchmarks of the Astral mass analyzer for glycoproteomics, it is hard to compare our data with previous studies.

Recently, a plasma proteomics study was published using narrow-window DIA on the Orbitrap Astral MS[32]. Although this study did not target glycopeptides and used a lower $MS^1$ precursor range (380−980 *m/z*), we were interested in comparing the proteins identified in that study to those identified using nGlycoDIA. At first glance, the here reported number of 181 detected glycoproteins may seem rather low, as in the previous mentioned more than 4500 proteins were identified. However, it should be noted that the latter study used an extracellular vesicle (EV)-based enrichment to reduce the dynamic range of the plasma sample. In that same study, 667 proteins were identified from "crude plasma" without EV-based enrichment, which is still about 3-fold more than we report. One would expect a priori that all 181 glycoproteins we identified here would also have been detected in this much "deeper" study, that used the narrow-window DIA method[32]. However, out of these 181 just 43 (24%) were also identified by Heil et al. in the crude plasma. Even in their dataset after EV-based enrichment (4500 identification) only 126 (70%) of the glycoproteins we identified are present, leaving 55 glycoproteins uniquely identified by the here described dedicated nGlycoDIA approach (30%). These 55 are almost all low abundant glycoproteins which have been described as genuine plasma proteins, often only detected by targeted highly sensitive enzyme-linked immunosorbent (*ELISA*) assays[42]. Their reported plasma concentrations range from 1 mg/L down to 1 ng/L. For most of these low abundant proteins, glycosylation has not been experimentally reported by MS-based plasma proteomics, although many of the here observed glycosites are annotated as putative *N*-glycosites in the public depositories UniProt or NextProt. Evidently, although ELISA assays can be very sensitive, they are not directly useful in detecting PTMs. Amongst these low abundant proteins were several glycosylated cytokines. This data will be valuable for understanding the role of glycosylation in the secretion and function of these cytokines.

One initial conclusion we can draw is that, although the absolute number of identified proteins in our nGlycoDIA proteomics approach is only a fraction of the ~4500 proteins reported in the recent nDIA EV-enriched plasma studies on the same Orbitrap Astral system, we were able to expose a unique part of the plasma proteome. The two-step enrichment (both offline using HILIC-based glycopeptide enrichment and in the gas-phase with the optimized $MS^1$ window) and the speed and sensitivity of the Orbitrap Astral MS and the here developed nGlycoDIA all aided to the expanded view of plasma glycoproteomics and the coverage of a broad dynamic range ($10^7$).

However, our study focused heavily on narrow-window DIA and the ability to analyze it using a DDA workflow. Broadening the window would allow for an increase in injection time, which might be advantageous for improving the data quality. However, as we demonstrated, the number of chimeric spectra increased substantially by extending the window size, while the overall Byonic scores remained similar. As our workflow relied primarily on the use of Byonic, we therefore decided to minimize the occurrence of chimeric spectra. However, even if chimeric spectra increased, the data analysis of wider windows was feasible, enabling the translation of this method to other instruments that require wider isolation windows. This has been demonstrated recently, when a similar Byonic strategy was used on DIA data with 16 Th windows[30]. However, this was performed on a purified protein, thus the data was much less complex, and the chance of chimeric spectra is lower than when a complex sample such as plasma is used. Ultimately, we decided to use narrow windows (3 Th) to reduce the possible occurrence of chimeric spectra. The software Byonic was chosen, because it was the only analysis software available that

accepted the DIA data format and was able to annotate the spectra as if it was DDA data, namely by assigning the $MS^1$ precursor masses and charge states. We attempted using MSFragger within FragPipe in a similar way (using the mass calibration option), however, this yielded only around 25 glycoPSMs. Other options we tried were DIA-Umpire, which led to a big decrease in unique glycoPSMs compared to Byonic, albeit with a 92% overlap in identifications, and the use of an experimental library-based algorithm in FragPipe (glyco-N-DIA), which yielded 61% overlap. Ultimately, library-free DIA is not yet well established for glycoproteomics, and the availability of such algorithms might require different experimental choices.

Furthermore, we used four different collision energies here, which each provided important information for compositional and structural annotation of the glycopeptides. Although an NCE of 35% led to the most identifications, the Y-ion coverage was lower compared to 30% NCE for most peptides. Therefore, we focused primarily on the 30% NCE data, or the data that was cumulatively identified. For example, the runs at NCEs of 25% and 35% could be used complimentarily to correlate glycan ions and backbone fragments, although this would still require the development of dedicated software tools. Being able to use stepping collision energy without sacrificing speed would enable all complementary information to be acquired in a single scan, as is often used in DDA glycoproteomics[14,15,38]. However, applying stepping HCD in this case meant we had to either reduce the cycle time, the injection time, or the MS1 scan range, or we had to increase the window size which would lead to more chimeric spectra. To stay as close to DDA type data and reduce chimeric spectra, we chose to work with a single collision energy for each run. Moreover, manual validation remained crucial for detailed glycan structure identification, as search engines only provide compositional information, and are still prone to incorrect structural annotations.

Lastly, our analysis focused mostly on qualitative analysis and the ability to detect glycopeptides from low abundant proteins, and not on the quantification of those glycopeptides even though this could be possible from integrating the $MS^1$ level; with the here-described method, $MS^1$ scans are recorded every second, leading to 5 points per peak in the shortest method (10-min, ~6 s peak width), which would allow for the integration. However, one advantage of DIA data would be the ability to perform quantification based on $MS^2$, as has previously been exploited by Yang et al. through their GproDIA algorithm[24]. In addition, with a suitable spectral library, quantification could be performed using DIA-NN, although the software itself does not support the generation of such a library. Counteracting this using an experimental workflow in FragPipe that does offer library generation, we did pilot a robust quantification between replicates and between the 10- and 40-min methods, but this should be further validated. It has to be noted that our library was generated from 4 DDA experiments without fractionation, and thus only included the higher abundant glycopeptides that could be identified in DDA.

Despite these limitations, the data produced by the Orbitrap Astral MS facilitated in-depth analysis of a large set of proteins in plasma. We showed that qualitative data independent acquisition from a single plasma sample can provide rich information on glycopeptide diversity. This aligns with our primary objective and underscores the potential for future research, exploring even more extensive glycoproteomic signatures and glycan structures on low abundant proteins.

Future studies could aim to improve software for library-free DIA analysis of glycopeptides, for example by implementing deep-learning for in silico-predicted library generation, similar to DIA-NN[20]. The elaborate dataset presented here could be used as training data for such applications. Furthermore, the ability to analyze glycan signatures on very low abundant proteins without immunoprecipitation or fractionation, opens the door for novel insights in disease mechanisms. Additionally, this method, and especially the ability to do in-depth glycoproteomics in only 10 min per sample, opens pathways for

broader applications, including clinical screenings, high-throughput analysis and biomedical research, where insights into glycan structures could inform disease diagnosis or treatment strategies. As a by-product of our work, we hypothesize that quantitative analysis of very low abundant cytokines in plasma may be helped by using glycopeptide enrichment, as several of the here detected cytokines are typically not observed in standard plasma proteomics.

Overall, this work establishes a good foundation for in-depth plasma glycoproteomics, where a simple and cost-effective enrichment followed by a short DIA method enables the identification of a broad range of glycoproteins and their glycans.

## Methods

### Sample preparation

Plasma samples were digested and enriched in 10 µL batches and are all derived from the same pooled plasma sample (VisuCon-F Normal Donor Set (EFNCP0125), Affinity Biologicals). 10 µL of pooled plasma sample (was mixed with 40 volumes of SDC-buffer (0.1M TRIS/HCl pH 8, 40 mM TCEP, 100 mM CAA, 1% SDC, $v/v$) for denaturation, reduction and alkylation. These samples were heated to 80 °C for 10 min, after which the proteases Trypsin and LysC were added in a 1:70 and 1:50 enzyme to protein ratio, respectively. Following overnight digestion at 37 °C, the samples were quenched with 0.1% TFA. The peptide digests were dried in vacuo and reconstituted in 0.1% TFA. Then, samples were loaded onto Oasis PRiME HLB (10 mg) 96-wells plates (Waters), washed with 0.1% TFA, and peptides were eluted using 60% ACN/0.1% TFA. The obtained digests were either analyzed directly by LC-MS/MS as the crude plasma sample or enriched for glycopeptides.

Glycopeptide enrichment was performed using cotton-HILIC SPE[57]. 100% cotton thread was cut into 10 mm lengths and individual threads were separated. Each thread was placed in a pipette tip, of which 8 at the time were used on a multichannel pipette. The tips were conditioned in 0.1% TFA (3 × 100 µL) and washed with 80% ACN/0.1% TFA (3 × 100 µL). Plasma peptide samples were reconstituted in 80% ACN/0.1% TFA (100 µL) and loaded onto the tips by 30 times repetitive pipetting. Tips were washed by three times repetitive pipetting in 80% ACN/0.1% TFA (3 × 100 µL), and finally eluted by three times repetitive pipetting in 50% ACN/0.1% TFA (3 × 100 µL). The three elution fractions were combined and dried in vacuo. Glycopeptide enriched samples were reconstituted in 100 µL 0.1% TFA and crude plasma samples were reconstituted in 1 mL 0.1% TFA for an approximately 200 ng/µL concentration. Peptide solutions were split in 12 µL batches in glass vials for LC-MS/MS analysis and 1 µL was injected per analysis.

### LC-MS/MS analysis

All data were acquired using a Thermo Scientific Vanquish™ Neo UHPLC (Thermo Fisher Scientific, Germering, Germany) coupled online to an Orbitrap Astral MS (Thermo Fisher Scientific, Bremen, Germany). The method order was randomized to reduce dilution effects, with the only condition that all enriched plasma samples were analyzed first to limit potential carry-over from the crude plasma samples. Peptides were separated on an IonOpticks Aurora Ultimate TS nanoflow UHPLC column (25 cm × 75 µm inner diameter, C18 stationary phase, 120 Å pore size, 1.7 µm particle size, IonOpticks PtyLtd, Fitzroy, Australia), in direct-injection configuration. The column was heated at 55 °C. The LC mobile phases used were water with 0.1% formic acid (solvent A) and 80% acetonitrile in water with 0.1% formic acid (solvent B) (both Optima LC/MS Grade, Fisher Chemical). Four different gradients were used: the concentration of solvent B was gradually increased from 1 to 4% B at a flowrate of 0.55 µL/min, to 8% B in 0.9 min at a reduced flowrate of 0.3 µL/min, and to 30%, either in 10, 20, 30 or 40 min, for the 10, 20, 30, and 40 min gradient runs, respectively. Lastly, the column was washed by increasing solvent B to 99% over 3 min and maintaining this concentration for 3 min, both at a flowrate of 0.55 µL/min.

The Orbitrap Astral mass spectrometer was operated in positive ion mode, with a spray voltage of 2000 V and an ion transfer tube temperature of 290 °C. Before submission of the sequence, the mass spectrometer was calibrated in positive ion mode for FTMS mass accuracy, Astral mass accuracy as well as ion foil and prism 2 for ion transmission. For the DDA measurements the MS$^1$ parameters were as follows: Orbitrap resolution: 120,000 (FWHM at $m/z$ 200); scan range: either a wide range of $m/z$ 300–2000 or a narrower range of $m/z$ 955–1655; maximum injection time: 50 ms; AGC target: 300%. The MS$^2$ parameters were as follows: intensity threshold: 5000; charge states: 2–8; dynamic exclusion: 7 s; isolation window: 1.6 or 3 Th; HCD collision energy: 30%; scan range: $m/z$ 150–2000; maximum injection time: 10 ms; AGC target: 200%. For the standard DIA method, the MS$^1$ settings were as follows: Orbitrap resolution: 120,000 (FWHM at $m/z$ 200); scan range: $m/z$ 380–980; maximum injection time: 100 ms; AGC target: 500%. The DIA parameters were as follows: isolation window: 2 Th; HCD collision energy: 27%; precursor mass range: $m/z$ 380–980; scan range: $m/z$ 150–2000; maximum injection time: 3 ms; AGC target: 200%. For the nGlycoDIA method, the MS$^1$ settings were as follows: Orbitrap resolution: 120,000 (FWHM at $m/z$ 200); scan range: $m/z$ 955–1655; maximum injection time: 100 ms; AGC target: 500%. The DIA parameters were as follows: isolation window: 3 Th; HCD collision energy: 25%, 27%, 30%, or 35%; precursor mass range: $m/z$ 955–1655; scan range: $m/z$ 150–2000; maximum injection time: 4 ms; AGC target: 800%. The wider-window methods had the same settings as nGlycoDIA, with the following changes: isolation window: 6 Th or 12 Th, and maximum injection time: 8 s or 16 s, respectively. For all DIA experiments a default charge state of two plus was used, and the DIA window placement optimization was used. The major experimental parameters used are summarized in Table 1.

Each experimental condition was performed with four technical replicates, leading to in total 160 nGlycoDIA experiments, 16 DDA experiments, and 12 DIA experiments with bigger window sizes.

### Data analysis

Raw data files were searched using PMI-Byonic (v 5.5.2, Protein Metrics). The protein database was a focused database consisting of plasma proteins (annotated to be in blood by using the Human Protein Atlas depository[42,43]) and common isoforms (with higher frequency

## Table 1 | Overview and comparison of key experimental parameters in the DDA, standard nDIA and nGlycoDIA methods

|  | DDA method | Standard nDIA method | nGlycoDIA |
|---|---|---|---|
| Orbitrap resolution | 240000 | 120000 | 120000 |
| MS$^1$ scan range (m/z) | 300–2000 | 380–980 | 955–1655 |
| MS$^2$ isolation window (Th) | 1.6 | 2 | 3 |
| MS$^2$ maximum injection time (ms) | 10 | 3 | 4 |
| MS$^2$ AGC target (%) | 200 | 200 | 800 |

Compared to the standard nDIA method, the optimized nGlycoDIA method differs primarily in the chosen MS$^1$ scan range. A slightly larger sliding window was chosen to accommodate also for this higher m/z range and the broader isotope distribution of the larger glycopeptides. Excluding the most abundant "high" m/z ins originating from non-modified peptides (e.g. originating from albumin) we also could extend the AGC target and injection time.

than 5%) as can be found in NextProt[58] (3107 entries), reverse decoys were added by Byonic. The rationale not to search against the entire human proteome database was to decrease search time. Fully specific digestion with at maximum two missed cleavages was allowed, and the precursor and fragment mass tolerance were set to 10 and 20 ppm, respectively. Carbamidomethylation of C was set as fixed modification, variable modifications included oxidation on M or W and pyroglutamic acid formation on protein and peptide N-terminal Q and E. A glycan list consisting of 279 glycans commonly observed in human proteins was used, as provided in the Supplementary information (Supplementary Data 1). In this table and in the text glycan compositions are abbreviated, where N or HexNAc is *N*-acetylhexosamine, H or Hex is hexose, F or dHex is fucose, P or Phos is phosphomannose, and S or NeuAc is *N*-acetylneuraminic acid/sialic acid. Each peptide was allowed a maximum of one variable modification and one glycan. Precursor and charge assignments were computed from the MS[1] data and a maximum of two precursors was allowed per MS2 scan.

As data analysis in glycoproteomics experiments is known to depend on software peculiarities, we additionally analyzed the data using the MSFragger (v 4.1), diaTracer (v 1.1.5) and DIA-NN (v 1.8.2beta8) tools within FragPipe (v 22.0). For the DIA-Umpire search, DIA Pseudo MS[2] was enabled with default parameters with the 'Mass Defect Filter' turned off. As database file we used the reviewed human UniProt fasta with isoforms (UP000005640). The precursor and fragment mass tolerance were set to 10 ppm and 20 ppm, respectively. We allowed strict cleavage of trypsin (KR) with maximum of 2 miscleavages. Included variable modifications were oxidation (M, max occurrence: 3), pyroglutamic acid formation (N-terminal Q, C, and E, max occurrence: 1) and a total of 3 modifications were allowed per peptide. "validation", "PTM-Shepherd" and "Glyco" workflows were enabled using default settings and employing the same glycan database as for the Byonic search. For library-based DIA, the default N-glyco-DIA workflow was selected with the same variable modification settings as for the DIA-Umpire search.

Subsequent data analysis was performed using R (v 4.3.1). FDR filtering of Byonic data was based on the individual PSM scores: PSMs were ordered from high to low score and a cut-off was calculated per file to have no more than 1% reverse sequences. This led to a run-specific score cut-off based on peptide FDR, the mean score cut-off being 167 (95% CI [163.8–176.0]). The exact score cut-offs can be found in Supplementary Data 2. After score filtering, data was additionally filtered on the criterium that the PSM had to be identified in at least two out of four replicates within any of the measurement conditions. The package ComplexHeatmap (v 2.20.0) was used for heatmaps and clustering, with confidence intervals being calculated using the rcompanion (v 2.4.36) package, other plots were made using ggplot2 (v 3.5.1), ggpubr (v 0.6.0), ggrepel (v 0.9.5), GGally (v 2.2.1), UpSetR (v 1.4.0), eulerr (v 7.0.2). Other used packages are: tidyverse (v 2.0.0), seqinr (v 4.2-36), openxlsx (v 4.2.5.2), scales (v 1.3.0). Reported (adjusted) p-values were calculated using ANOVA, followed by the Tukey HSD test. Ion coverage calculations were performed using the in-house developed multi-annotator using Rust and rustyms (available on GitHub: https://github.com/snijderlab/annotator and https://github.com/snijderlab/rustyms, multi-annotator and rustyms, respectively)[59]. The R-script will be available as Supplementary Data 3, and the multi-annotator version used here is available as Supplementary Software.

**Structural glycan modeling**
Structural modeling of the most frequently identified glycan composition on the cytokines IL-12, IL-22, and VIP was done using GLYCAM-Web[60]. The structural glycan features such as fucose position and bisecting GlcNAc were determined manually by assigning the fragmentation spectra, additional structural elements were inferred from the literature[6,50]. The modeled glycans were: NeuAc-α2,6-Gal-β1,4-GlcNAc-β1,2-Man-α1,3-(Man-α1,6-(Man-α1,3)-Man-α1,6-)Man-β1,4-GlcNAc-β1,4-GlcNAc-β1-Asn for N3H6S1, NeuAc-α2,6-Gal-β1,4-GlcNAc-β1,2-Man-α1,3-(Gal-β1,4-GlcNAc-β1,2-Man-α1,6-) Man-β1,4-GlcNAc-β1,4-GlcNAc-β1-Asn for N4H5S1, NeuAc-α2,6-Gal-β1,4-GlcNAc-β1,2-Man-α1,6(NeuAc-α2,6-Gal-β1,4-GlcNAc-β1,4(NeuAc-α2,6-Galβ1,4-GlcNAc-β1,2-)Man-α1,3-)Man-β1,4-GlcNAc-β1,4-GlcNAc-β1-Asn for N5H6S3.

**Reporting summary**
Further information on research design is available in the Nature Portfolio Reporting Summary linked to this article.

## Data availability
The raw LC-MS/MS data files and Byonic output files generated in this study have been deposited in the MASSive repository under accession code MSV000095471[61] [https://massive.ucsd.edu/ProteoSAFe/dataset.jsp?task=628297fa791f4a27bce5d1a7885fe347] and in the ProteomeXchange repository under accession code PXD054333. The data from the Human Protein Atlas[42,43] was used to infer blood concentration of the proteins identified, which can be accessed here [https://www.proteinatlas.org/download/proteinatlas.tsv.zip]. Source data are provided with this paper.

## Code availability
The R-script used to filter, analyze and interpret the Byonic output files is available as Supplementary Data 3. The multi-annotator tool[59] used for ion-coverage calculation is available on Github [https://github.com/snijderlab/annotator] and the version used in this manuscript is available as Supplementary Software 1.

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

## Acknowledgements

This research received funding through the Dutch Research Council (NWO) funding the X-omics Road Map program (project 184.034.019). A.J.R.H. acknowledges support from NWO through the Spinoza Award SPI.2017.028.

## Author contributions

S.J., K.R.R., A.A.M. and A.J.R.H. conceptualized the project; S.J., M.Z. and A.P. performed the experiments; S.J., M.Z., A.P. and E.D. worked on method development; S.J. and D.S. performed data analysis and visualization; A.A.M. and A.J.R.H. provided resources; S.J. and A.J.R.H. were responsible for the writing; all authors further edited the work and critically revised the final version of the manuscript.

## Competing interests

The authors declare the following potential competing interests: M.Z., A.P., E.D. and A.A.M. are employees of Thermo Fisher Scientific, the manufacturer of the mass spectrometers used in this study. The remaining authors declare no competing interests.
