## [Transparent Peer Review file · Nature Communications]

In-depth plasma N-glycoproteome profiling using narrow-window data-independent acquisition on the Orbitrap Astral Mass Spectrometer

Corresponding Author: Professor Albert Heck

Version 0:

Reviewer comments:

Reviewer #1

(Remarks to the Author)

The manuscript presents significant advancements in glycoproteomics by leveraging the high speed, resolution, and sensitivity of the Orbitrap Astral mass spectrometer, allowing for deep glycoprotein coverage within significantly shorter analysis times. The authors provide strong evidence that the narrow-window data-independent acquisition (nDIA) strategy enables the identification of over 3,000 unique glycopeptide-spectrum matches (glycoPSMs) from 181 glycoproteins—a substantial achievement for high-throughput glycoproteomics. However, several key aspects of the data analysis workflow, particularly the use of Byonic software for glycopeptide identification, require greater scrutiny and transparency.

Strengths:

- o The mass spectrometer's ability to identify glycoproteins over a dynamic range of 7 orders of magnitude within 40 minutes is a notable advancement, particularly for clinical and high-throughput applications, where both speed and sensitivity are critical. The authors also report identifying over 2,500 glycoPSMs using only a 10-minute LC gradient, highlighting the system's potential for high-throughput glycoproteomics in large-scale clinical studies.

- o The nGlycoDIA approach provides clear advantages in terms of reproducibility and sensitivity, particularly when paired with cotton-HILIC glycopeptide enrichment. The manuscript effectively demonstrates this through robust comparisons of glycoprotein identification across different LC gradient times and collision energies.

- o The identification of more than 80 distinct glycans on a single site (N639 of serotransferrin) illustrates the depth of analysis possible with this method, further underscoring the capabilities of nGlycoDIA in providing detailed glycosylation profiles.

Weaknesses:

- o The manuscript heavily relies on Byonic software for data analysis, which is a widely used tool for glycopeptide identification but is known to sometimes suffer from high false discovery rates (FDR), especially when analyzing complex samples like plasma. The manuscript mentions applying a 1% FDR using reverse decoy sequences, but it does not provide critical details on the specific Byonic score thresholds used for glycopeptide identification or how FDR was controlled beyond this. Without explicit information on these cut-offs, it is difficult to fully assess the reliability of the glycopeptide identifications. The authors should clearly state the Byonic score thresholds used and provide more transparency on how manual validation or cross-validation with other software tools was conducted. This is essential because without stringent filtering, Byonic results can lead to a significant number of false positives, undermining the overall confidence in the findings. Further, including a discussion on alternative approaches, such as using additional software for cross-verification (e.g., MSFragger or FragPipe), could enhance confidence in the data interpretation. This would not only increase the robustness of the conclusions but also provide readers with a clearer understanding of the measures taken to mitigate false discoveries.

- o The manuscript introduces structural modeling of two glycans (N3H6S1 and N5H6S3) but does not explain why these particular glycans were chosen. The authors should clarify whether these glycans are representative of the broader glycosylation patterns or were selected based on abundance or biological significance. Moreover, the manuscript lacks details on how glycopeptide structures were confirmed beyond Byonic assignments. Were any manual inspections conducted, or were orthogonal methods (e.g., targeted MS/MS) employed to validate key identifications? Including this information is crucial to ensure confidence in the biological relevance of the findings.

- o The decision to use a 10 ppm precursor mass tolerance is questionable, considering the higher mass accuracy offered by the Orbitrap Astral MS. The authors should justify why this tolerance was chosen and explore whether using narrower mass tolerances could improve glycopeptide identification specificity, particularly in complex samples.

- o The manuscript does not clarify why different LC-MS/MS columns were used for the DDA and DIA analyses, raising

concerns about the comparability of results across both methods. The authors should explain the rationale behind this decision and discuss whether using the same column would produce more consistent datasets.

o The manuscript lacks clear reporting on the data cut-offs used to filter out low-confidence glycopeptide identifications. Specifically, the authors need to state the score threshold or FDR applied during the analysis to ensure that the reported identifications are reliable.

o The supplementary files are not listed in a logical or chronological order, making it difficult to locate data corresponding to specific figures or sections. Reorganizing these files and ensuring they are properly referenced in the manuscript would enhance readability and flow.

o The claim that "DIA is rapidly becoming the new standard for proteomics" is overly bold and lacks sufficient scientific support. While DIA is gaining prominence, particularly for high-throughput, quantitative proteomics, it has not fully replaced DDA, which remains essential for discovery-based research. A more accurate statement would acknowledge that DIA is increasingly adopted in applications where reproducibility and throughput are critical, while DDA continues to play a key role in exploratory studies.

o The manuscript claims that shortening the LC gradient to 10 minutes still allows for the detection of almost 2,500 unique glycoPSMs. However, this claim would benefit from statistical support, such as uncertainty bars or confidence intervals, to illustrate the variability of the data. Providing such statistical measures would help readers assess the robustness of the method.

o Figure 5 is missing uncertainty bars in its bar graphs, which are important for representing variability and reproducibility. Given that glycoproteomics data can often be noisy, adding uncertainty bars would provide a clearer picture of the confidence in the reported identifications.

(Remarks on code availability)

Please see comments to the author

Reviewer #2

(Remarks to the Author)

In this research article, Jager et al. describe a narrow window DIA method (nGlycoDIA) for the identification of glycopeptides in plasma. The narrow windows are enabled by the high scan speed of the Astral analyzer. The data were searched using conventional DDA software for glycoproteomics (Byonic). The authors report the identification of low-abundance proteins, particularly cytokines, and describe glycosylation sites that have not been experimentally reported before. This paper demonstrates the potential of Astral for glycoproteomics and could be particularly valuable for researchers using this platform. However, the results are difficult to contextualize as they are not benchmarked against conventional DDA methods commonly used in the field and they base their analysis on a single DDA-based software without glycan FDR control. Further, they don't describe how this approach performs for quantitative experiments, which limits its applicability.

Major Comments:

-It is challenging to assess the results without a direct comparison to conventional DDA methods. The authors primarily compare nGlycoDIA to a "standard DIA method." For most readers, it would likely be more informative to see how it compares to standard DDA in terms of identifications.

-Although the narrow windows indeed make DIA and DDA methods similar, the authors use a 3-Da window compared to the 1.6 Da used in the DDA approach. How does this affect the confidence of the identifications? At what point would a DDA-based analysis of DIA data no longer be reliable? Would it be feasible to run a DIA method with wider windows (e.g., on Exploris) to demonstrate the benefits of narrow windows and the challenges associated with wider windows? How do these factors impact the confidence of the identifications?

-The comparison of gradient lengths is interesting, but it seems the authors used the same MS methods for different gradients. What was the duty cycle, and was it adjusted to accommodate the gradient length/peak width? One advantage of longer gradients is the potential for increased duty cycles and applying smaller windows. Was the injection amount adjusted, and could more sample be loaded for the longer gradients (assuming the columns weren't overloaded)? How many points per peak were obtained, and how would this affect quantification?

-One key advantage of using DIA is the ability to perform (label-free) quantification, especially using MS2 data. Could this method be applied to a quantitative experiment? Which ions would be selected for quantification? If possible, the authors could even perform a small quantitative benchmark experiment, especially in testing if glycan Y-ions (more prevalent in low collision energy) are consistently detected for quantification, similar to the principle of GproDIA (Yang et al Nature Comm 2021).

-The authors report an increase in glycopeptide identifications when the collision energy is increased from 25% to 35%. However, as noted, the quality of PSMs decreases with this increase. Since fewer glycan Y-ions are identified at higher NCE, how confident can they be in these glyco-PSMs, especially since Byonic's FDR calculation is primarily based on peptides?

-A significant concern is the reliance on a single software solution for identifications, as it has been shown that results strongly depend on the software and settings used (Kawahara et al., 2020, Nature Methods). Thus, it is difficult to judge how confident the reported identifications are. In principle, the data should be analyzable with any DDA-based glyco software.

Could the authors analyze the data using MSFragger (Polasky et al., 2020, Nature Methods) or pglyco3 (Zeng et al, Nature Methods 2021) to assess the overlap in identifications.

Minor Comments:

-Figures are not in the correct order.

-Figure 3 is incorrectly referenced in the text ("Figure 3D and 3E, Suppl...")—it should be Figure 2.

-Figure 6A is incorrectly referenced—it should be Figure 5A.

-Optional minor Comment: The authors mention an increase in unique PSMs for non-glycopeptides in enriched plasma compared to crude plasma. This is an interesting finding. What are the additional proteins they identified?

(Remarks on code availability)

Reviewer #3

(Remarks to the Author)

(Remarks on code availability)

Reviewer #4

(Remarks to the Author)

In this manuscript, the authors utilized the latest Astral instrument with narrow-window DIA for plasma glycopeptide identification, which represents a new application of the Astral platform. However, I believe that this paper mainly presents one application of the Astral narrow-window DIA without addressing the critical challenges in glycopeptide DIA identification. It lacks systematic optimization and comprehensive comparison.

1. One of the key advantages of DIA analysis is its ability to avoid the random selection of precursor ions in DDA. However, since DIA captures all MS2 data within a given window, a significant challenge lies in building the spectral library and ensuring precise matching and interpretation. While narrow-window DIA theoretically reduces chimeric spectra, the manuscript does not provide any validation of its accuracy, nor does it compare the results to existing glycopeptide DIA analyses or DDA results.

2. Another key advantage of DIA is the ability to perform simultaneous quantification alongside identification. However, the authors did not present any quantitation results. Considering the identification level in this work does not seem to surpass current glycopeptide identification techniques significantly, the quantitative results and their accuracy should also be demonstrated.

3. As a novel application for glycopeptide analysis, systematic optimization of the Astral instrument parameters is crucial. Although the authors mention some parameters, such as "a downside of shifting the MS1 range," I regret to note that beyond exploring different fragmentation energies (which is achievable on any Orbitrap instrument post-Fusion), the manuscript does not show results from other systematic parameter optimizations.

4. After applying the 950-1655 window, the authors mention a reduction in non-glycopeptide identifications. What about the number of glycopeptide identifications?

5. Why did the authors choose a library annotated for blood proteins instead of a comprehensive human protein library?

6. Page 8: "Almost all identified glycosylated PSMs found..."—What does "almost all" mean? 80%? 90%? This should be quantified.

7. Page 8: "..., indicating high method robustness (Figure 3D and 3E, Supplemental Figure S4A and S4B)" should refer to Figures 2D and 2E.

8. In "Setting up narrow-window nGlycoDIA," it states 955-1650, while Table 1 and the Supplementary Figure S1 show 950-1655. These discrepancies should be corrected.

9. The background does not sufficiently summarize the current state of glycopeptide DIA analysis. Key works such as GproDIA and GlycoDIA are missing and should be cited.

(Remarks on code availability)

I did not run the entire dataset to check for reproducibility and only tested a small subset of the data. The code includes a README file with sufficient instructions for installation and running.

Version 1:

Reviewer comments:

Reviewer #1

(Remarks to the Author)

have carefully reviewed your rebuttal and the corresponding revisions made to the manuscript. I appreciate the effort you

have put into addressing all the comments and concerns raised during the review process. Your thorough and well-structured responses have significantly improved the clarity and rigor of the manuscript.

The modifications you have implemented enhance the overall quality of the study, ensuring that your findings are well-supported and effectively communicated. I am satisfied with the revisions and have no further concerns. The manuscript is now suitable for publication in its current form.

Thank you for your diligence in refining the work. I look forward to seeing your study published.

(Remarks on code availability)

Reviewer #2

(Remarks to the Author)

The authors have addressed the reviewers' comments and improved the manuscript. I recommend acceptance of this work but have one minor comment that arose from the revision, which could be incorporated into the discussion: it is interesting to note that the identifications increased with window size. Are these results reliable, and does this suggest that this approach could also be adapted to other (older) MS platforms that typically require wider windows? Including a discussion on this point could be valuable, as it would emphasize that this approach is not limited to users of the Astral platform.

(Remarks on code availability)

Reviewer #3

(Remarks to the Author)

(Remarks on code availability)

Reviewer #4

(Remarks to the Author)

The authors have addressed most of my concerns. The revisions significantly strengthen the manuscript, and the improvements in clarity and data presentation are highly appreciated.

The additional analysis of chimeric spectra and comparisons across different isolation windows, coupled with cross-validation using FragPipe, effectively address the concern regarding the accuracy of the DIA method. The authors' efforts to minimize chimeric spectra by optimizing window size and injection time are commendable. The clear presentation of results, such as achieving only 2.5% chimeric spectra at a 3 Th window, inspires confidence in the methodology. I have no further concerns regarding this point.

The authors have also provided a reasonable explanation for the lack of direct quantification results, acknowledging the current limitations of available software for DIA glycopeptide quantification. Furthermore, the rationale for using a blood protein-specific library to reduce search space and improve efficiency is acceptable. I believe the limitation reflects the broader challenges in developing tools tailored for DIA glycopeptide analysis, which applies not only to this study but to the field as a whole. The authors should discuss this issue in the manuscript.

Regarding the identification numbers, it is undeniable that this work does not represent the forefront in terms of quantity, as many glycoproteomics studies report higher identification counts. While I do think the focus of glycoproteomics is shifting away from merely pursuing higher identification numbers, the authors should discuss this point in the manuscript, as the primary goal of DIA analysis is to enable deeper and broader insights into glycopeptide characterization.

(Remarks on code availability)

I did not reviewed the code in this revision.

Detailed responses to the Reviewers (comments are in “normal” our responses are in bold)

Reviewer #1 (Remarks to the Author):

The manuscript presents significant advancements in glycoproteomics by leveraging the high speed, resolution, and sensitivity of the Orbitrap Astral mass spectrometer, allowing for deep glycoprotein coverage within significantly shorter analysis times. The authors provide strong evidence that the narrow-window data-independent acquisition (nDIA) strategy enables the identification of over 3,000 unique glycopeptide-spectrum matches (glycoPSMs) from 181 glycoproteins—a substantial achievement for high-throughput glycoproteomics. However, several key aspects of the data analysis workflow, particularly the use of Byonic software for glycopeptide identification, require greater scrutiny and transparency.

Strengths:

- o The mass spectrometer’s ability to identify glycoproteins over a dynamic range of 7 orders of magnitude within 40 minutes is a notable advancement, particularly for clinical and high-throughput applications, where both speed and sensitivity are critical. The authors also report identifying over 2,500 glycoPSMs using only a 10-minute LC gradient, highlighting the system's potential for high-throughput glycoproteomics in large-scale clinical studies.

- o The nGlycoDIA approach provides clear advantages in terms of reproducibility and sensitivity, particularly when paired with cotton-HILIC glycopeptide enrichment. The manuscript effectively demonstrates this through robust comparisons of glycoprotein identification across different LC gradient times and collision energies.

- o The identification of more than 80 distinct glycans on a single site (N639 of serotransferrin) illustrates the depth of analysis possible with this method, further underscoring the capabilities of nGlycoDIA in providing detailed glycosylation profiles.

Weaknesses:

- o The manuscript heavily relies on Byonic software for data analysis, which is a widely used tool for glycopeptide identification but is known to sometimes suffer from high false discovery rates (FDR), especially when analyzing complex samples like plasma. The manuscript mentions applying a 1% FDR using reverse decoy sequences, but it does not provide critical details on the specific Byonic score thresholds used for glycopeptide identification or how FDR was controlled beyond this. Without explicit information on these cut-offs, it is difficult to fully assess the reliability of the glycopeptide identifications. The authors should clearly state the Byonic score thresholds used and provide more transparency on how manual validation or cross-validation with other software tools was conducted. This is essential because without stringent filtering, Byonic results can lead to a significant number of false positives, undermining the overall confidence in the findings. Further, including a discussion on alternative approaches, such as using additional software for cross-verification (e.g., MSFragger or FragPipe), could enhance confidence in the data interpretation. This would not only increase the robustness of the conclusions but also provide readers with a clearer understanding of the measures taken to mitigate false discoveries.

We thank the reviewer for the overall very positive and constructive comments. The reason we chose to use Byonic is because it is still the most well-established data analysis tool for glycoproteomics, but also because it can look back into the MS1 information to generate the precursor mass, which is one of the key requirements that enable the analysis of nGlycoDIA data. We agree, and are also critical on the FDR, or ill-definition thereof, in Byonic. Because the default FDR in Byonic is based on proteins, we felt this is not fully applicable when peptide-based enrichment is performed, or when the precursor range is limited (as is the case with the 955-1655 m/z MS¹ range). This removes many peptides and leads to the false annotation of (reverse) proteins with a single peptide annotated. Therefore, we based the FDR on reverse peptides (which can also have assigned glycans) and their score as provided by Byonic (which is based on both the peptide and glycan). All low-scoring PSMs are removed until only 1% of reverse peptides are left, leading to a 1% FDR and a different Byonic score based cut-off value for each run. To address the reviewers' point, we added these cut-off values to the supplemental information in Supplemental Table 2, the mean score cut-off being 167 (95% CI [163.8-176.0]). Furthermore, we filtered our data on the criterion that a PSM had to be identified in 2 or more injection replicates in at least one condition.

Additionally, as requested we performed extra data analysis using a different software suite, namely FragPipe/MSFragger and compared this data with the Byonic results. We dedicated an entire new section to this in the revised manuscript ("Compatibility with alternative data analysis software" on page 17 and added a new Supplemental Figure S9). In short, the current version available of MSFragger was not able to extract the precursor masses from the MS1 data. Therefore, we used DIA-Umpire (with the default settings) and the experimental N-glyco DIA algorithm in FragPipe, which is library based. We see high overlap between the identifications by MSFragger and Byonic, especially when using DIA-UMPIRE, although Byonic identifies substantially more glycopeptides. We expected that the Umpire strategy would lead to less identifications, because the Umpire algorithm relies on a high correlation of intensities in MS¹ and MS², across multiple scans. (Abundant) fragments that do not follow the general MS¹ peak shape disrupt this correlation, which can be the case with glycopeptides where fragmentation can change due to glycan isomers.

The experimental N-glyco DIA workflow in FragPipe did provide more identifications than DIA-Umpire (and subsequent quantification using DIA-NN), but this algorithm is not yet peer-reviewed. Also, not all glycans in our library could be used in the library generation step, resulting in around 1700 PSMs being ignored. Therefore, we see a smaller overlap between this method and our Byonic method. But, although the N-glyco-DIA workflow in FragPipe, did not lead to a high number of glycopeptide identifications, it allowed us to assess glycopeptide quantifications, which we now also discuss in S9 and the revised manuscript.

o The manuscript introduces structural modeling of two glycans (N3H6S1 and N5H6S3) but does not explain why these particular glycans were chosen. The authors should clarify whether these glycans are representative of the broader glycosylation patterns or were selected based on abundance or biological significance. Moreover, the manuscript lacks details on how glycopeptide structures were confirmed beyond Byonic assignments. Were any manual inspections conducted, or were orthogonal methods (e.g., targeted MS/MS) employed to validate key identifications? Including this information is crucial to ensure confidence in the biological relevance of the findings.

We appreciate this comment of the reviewer, because most of the work is based on glycopeptide identification at a compositional level. In Figure 5B and C, the glycan N4H5S1 was modeled on IL12A – N217, and the glycan N3H6S1 was modeled on IL12B – N135. These glycans were chosen because they were the most frequently identified glycans. We make this now more clear in the text: “Structural modeling of the most frequently identified glycan composition” – page 5, and “Furthermore, to demonstrate the relevance of these glycans, we modeled the most frequently identified glycan composition on structural models of the proteins, when available. – page 13 and shown in Figure 5A.

Validation of the composition and structural elements was performed manually, and no orthogonal methods were used. First, the proposed composition of the modelled glycans were validated by assigning the Y- and B-ions, as described in the text in detail as well (examples: “One of the glycoforms was consistently annotated as N4H5S2, however, after manual curation, we corrected this to N4H5F2S1” – page 13) and if data was sufficient, the glycan structural elements (fucose location and bisection) were assigned manually as well. All other assumptions on structure were inferred from literature, as was the case for the structures that we modeled. We understand that this was unclear from the initial manuscript and clarified this in the methods: “The structural glycan features such as fucose position and bisecting GlcNAc were determined manually by assigning the fragmentation spectra, additional structural elements were inferred from the literature” – page 5.

o The decision to use a 10 ppm precursor mass tolerance is questionable, considering the higher mass accuracy offered by the Orbitrap Astral MS. The authors should justify why this tolerance was chosen and explore whether using narrower mass tolerances could improve glycopeptide identification specificity, particularly in complex samples.

10 ppm is the default parameter in Byonic, which is why we selected this value. We examined the mass errors of the PSMs of the filtered Byonic output, which shows that, indeed, most precursors have (as expected) an error smaller than 10 ppm, as depicted in the histogram below where we plotted the precursor ppm errors of the 40-minute methods with NCE30 on enriched plasma. Thus, indeed, the ppm error can be lowered to a value such as 5 ppm. Only a minimal proportion of PSMs falls outside of the 5 ppm error range, and the errors are normally distributed, therefore we feel we do not need to re-analyze the data.

o The manuscript does not clarify why different LC-MS/MS columns were used for the DDA and DIA analyses, raising concerns about the comparability of results across both methods. The authors should explain the rationale behind this decision and discuss whether using the same column would produce more consistent datasets.

Initially, the DDA analysis was only performed to determine the optimal precursor mass range, which should not be influenced by the chromatography. Initially we restrained from a direct comparison between DDA and DIA, because we felt it would be hard to make a fair comparison. However, since the reviewers asked us to compare the DIA set to a DDA set, we have now added to the revised manuscript data from new experiments, including a dataset obtained from DDA analyses on the Orbitrap Astral using the same column and gradient as used for the DIA analyses, to make the comparison hopefully as fair as possible. These new analyses confirm our initial statement that nGlycoDIA provides a substantial benefit for identifying more (and less abundant) glycopeptides in plasma, as requested by the reviewers.

o The manuscript lacks clear reporting on the data cut-offs used to filter out low-confidence glycopeptide identifications. Specifically, the authors need to state the score threshold or FDR applied during the analysis to ensure that the reported identifications are reliable.

We added a table with the score cut-offs as supplementary data file (Supplemental Table 2). See also our responses about the FDR above.

o The supplementary files are not listed in a logical or chronological order, making it difficult to locate data corresponding to specific figures or sections. Reorganizing these files and ensuring they are properly referenced in the manuscript would enhance readability and flow.

We thank the reviewer for this comment, it was unfortunately overlooked by us. To enhance readability and flow in the revised manuscript we removed redundant supplementary figures (S2 and S4). We ensured all figures were in chronological order in the revision.

o The claim that "DIA is rapidly becoming the new standard for proteomics" is overly bold and lacks sufficient scientific support. While DIA is gaining prominence, particularly for high-throughput,

quantitative proteomics, it has not fully replaced DDA, which remains essential for discovery-based research. A more accurate statement would acknowledge that DIA is increasingly adopted in applications where reproducibility and throughput are critical, while DDA continues to play a key role in exploratory studies.

We agree that we may have been a bit bold here and have replaced that sentence on p. 2 with “While the use of DIA is gaining prominence, particularly for high-throughput, quantitative proteomics, it is not yet prevalent for discovery-based research and glycoproteomics”

o The manuscript claims that shortening the LC gradient to 10 minutes still allows for the detection of almost 2,500 unique glycoPSMs. However, this claim would benefit from statistical support, such as uncertainty bars or confidence intervals, to illustrate the variability of the data. Providing such statistical measures would help readers assess the robustness of the method.

The 2500 unique glycoPSMs mentioned in the abstract were the cumulative unique glycoPSMs found in 4 replicates (filtered for identification in at least 2 out of 4). We like the suggestion to use confidence intervals to describe the variability, therefore we changed this to the mean number of unique glycoPSMs (after filtering) with the 95 % confidence interval: 1850 (95% CI [1840-1860]). We did the same in the discussion on page 17 for the nGlycoDIA results of 40 minutes: for crude plasma in 30 minutes: 400 (95% CI [364-405], 30 minutes using NCE 30); and enriched plasma in 40 minutes: 3000 (95% CI [2940-3130]).

o Figure 5 is missing uncertainty bars in its bar graphs, which are important for representing variability and reproducibility. Given that glycoproteomics data can often be noisy, adding uncertainty bars would provide a clearer picture of the confidence in the reported identifications.

Figure 5A represents the total number of PSMs identified for the low abundant cytokines across all replicates and NCEs, and it serves to demonstrate the cumulative and repetitive evidence found for these glycoforms, as described in the legend. It does not serve to quantify the glycoforms, but just to show that they are present. Therefore, in this figure we cannot depict uncertainty bars.

Reviewer #2 (Remarks to the Author):

In this research article, Jager et al. describe a narrow window DIA method (nGlycoDIA) for the identification of glycopeptides in plasma. The narrow windows are enabled by the high scan speed of the Astral analyzer. The data were searched using conventional DDA software for glycoproteomics (Byonic). The authors report the identification of low-abundance proteins, particularly cytokines, and describe glycosylation sites that have not been experimentally reported before. This paper demonstrates the potential of Astral for glycoproteomics and could be particularly valuable for researchers using this platform. However, the results are difficult to contextualize as they are not benchmarked against conventional DDA methods commonly used in the field and they base their analysis on a single DDA-based software without glycan FDR control. Further, they don't describe how this approach performs for quantitative experiments, which limits its applicability.

We thank the reviewer for the positive comments on our study. In the revised manuscript we address the major concerns especially about the comparison with DDA and the use of alternative software, and the quantification, as described in more detail below.

Major Comments:

-It is challenging to assess the results without a direct comparison to conventional DDA methods. The authors primarily compare nGlycoDIA to a “standard DIA method.” For most readers, it would likely be more informative to see how it compares to standard DDA in terms of identifications.

We thank the reviewer for this comment, although comparing such different analyses approaches is always challenging, as they do require different optimal settings. To still make the comparison, we have now included in the revised manuscript analyses based on a series of DDA experiments on enriched plasma. To make a reasonable fair comparison with the DIA data, we have used for the DDA experiments a single collision energy (30% NCE) and the same column and gradient conditions. The results are now described on p 5-6, and Supplementary Figure S1. In general, we see that we identify 1903 (median) unique glycopeptides per injection. Around 95 glycoproteins were detected per injection with at least 2 glycoPSMs, and about 160 glycosites per injection, also with at least 2 glycoPSMs.

-Although the narrow windows indeed make DIA and DDA methods similar, the authors use a 3-Da window compared to the 1.6 Da used in the DDA approach. How does this affect the confidence of the identifications? At what point would a DDA-based analysis of DIA data no longer be reliable? Would it be feasible to run a DIA method with wider windows (e.g., on Exploris) to demonstrate the benefits of narrow windows and the challenges associated with wider windows? How do these factors impact the confidence of the identifications?

To address this interesting comment, we explored the 3 Th window in DDA, which showed that the identified peptides overlap for more than 85% between 1.6 Th and 3 Th isolation windows (about the same overlap as between injection replicates), which shows that Byonic is still providing consistent identifications. Furthermore, we tested larger windows in DIA. Here, we found that the occurrence of chimeric spectra increases quickly. Because DDA software is not optimal for the interpretation of chimeric spectra, we felt this would not be as reliable. Using library-based DIA software the conclusion might be different, but this would require glycopeptide-based libraries that are beyond the scope of this work. We report the results in detail on page 6. In summary, we see that the glycopeptides found are highly identical (85%) between 1.6 dan 3 Th, and that beyond 3 Th, the occurrence of chimeric spectra increases a lot.

-The comparison of gradient lengths is interesting, but it seems the authors used the same MS methods for different gradients. What was the duty cycle, and was it adjusted to accommodate the gradient length/peak width? One advantage of longer gradients is the potential for increased duty cycles and applying smaller windows. Was the injection amount adjusted, and could more sample be loaded for the longer gradients (assuming the columns weren't overloaded)? How many points per peak were obtained, and how would this affect quantification?

We thank the reviewer for this valid comment. The cycle time of the method was 940 ms, which we intentionally kept like the narrow window method benchmarked on the Orbitrap Astral by Heil et al (JPR 2023) and Guzman et al (Nat Biotech 2024), which we deemed 'standard nDIA'. We have added this information to the revised manuscript ("In previously published ... standard nDIA method" – page 6). Standard nDIA was benchmarked with similar LC-conditions (length and gradient) as our 10-min method, with 6 s peak width. This would lead to 5 points per peak. This typically allows for quantification based on both MS1 and MS2. Because our method was based on the shortest method, we did not change it for the longer ones.

-One key advantage of using DIA is the ability to perform (label-free) quantification, especially using MS2 data. Could this method be applied to a quantitative experiment? Which ions would be selected for quantification? If possible, the authors could even perform a small quantitative benchmark experiment, especially in testing if glycan Y-ions (more prevalent in low collision energy) are consistently detected for quantification, similar to the principle of GproDIA (Yang et al Nature Comm 2021).

As suggested by the reviewer, we performed efforts to quantify the glycopeptides. We tested MS2 based quantification using the experimental workflow in MSFragger (Fragpipe v 22.0, MSFragger v 4.1) (kindly provided by the Nesvizhskii group), which can generate spectral libraries from DDA data, and quantified the DIA data using DIA-NN. However, this algorithm is not published and peer reviewed yet. This test led to highly reproducible quant values for both the 10-minute run, and the 40-minute run. The drawback here is that the used glycopeptide library did not contain any of the lower abundant peptides we detected here, since it was based on the DDA data, therefore we do not know how reliable the quantification is for lower abundant glycopeptides. We discuss this issue in detail on page 19 ("An interesting feature...most abundant glycopeptides") and 21 ("Lastly, our analysis...identified in DDA"), and in the new Supplemental Figure S9.

-The authors report an increase in glycopeptide identifications when the collision energy is increased from 25% to 35%. However, as noted, the quality of PSMs decreases with this increase. Since fewer glycan Y-ions are identified at higher NCE, how confident can they be in these glyco-PSMs, especially since Byonic's FDR calculation is primarily based on peptides?

We completely agree with the reviewer that Y-ions are crucial for confident glycopeptide assignment. Therefore, we focus mostly on the results with NCE 30, which had lower backbone fragmentation, but better Y-ion coverage, as can be seen in Supplemental Figure S3C. Additionally we stress that multiple collision energies would be beneficial for confident glycopeptide assignment. Since this is an important consideration, we decided to further expand on this in the discussion on page 21 ("Furthermore, we used...incorrect structural annotations").

-A significant concern is the reliance on a single software solution for identifications, as it has been shown that results strongly depend on the software and settings used (Kawahara et al., 2020, Nature Methods). Thus, it is difficult to judge how confident the reported identifications are. In principle, the data should be analyzable with any DDA-based glyco software. Could the authors analyze the data using MSFragger

(Polasky et al., 2020, Nature Methods) or pglyco3 (Zeng et al, Nature Methods 2021) to assess the overlap in identifications.

We agree with the reviewer that the choice of software can have a substantial impact on the reported PSMs and that methods should be as replicable and cross-platform as possible. To address this concern, we have now also processed our data in Fragpipe (v. 22.0) using MSFragger (v 4.1), the experimental library-based DIA workflow in Fragpipe (v. 22.0), as well as DIA-Umpire. From this, we could see that both Fragpipe based workflows yield lower coverage of the plasma glycoproteome, compared to Byonic, as we expected. DIA-Umpire relies on consistent, high-correlation feature detection between MS¹ and MS² signal, glycopeptides often do not elute in a gaussian peak, and peak splitting is often observed due to isomeric differences in the glycans (mostly with sialic acid linkages, that are very abundant in plasma), which influences their fragmentation as well. This disrupts the correlation between MS¹ and MS² features, reducing the number of matches precursors. The glycopeptides that were identified did have a very high overlap with the ones identified in the Byonic searches (92%). With the library-based search, a library was generated from the DDA data, thus the number of identifications could not exceed that. Here, about 1700 PSMs could not be incorporated into the library, because they did not have a UniMod annotation, effectively decreasing the glycan library. Thus, only 61% of identifications (~1000 unique glyco-PSMs) were also found with our Byonic search, and 300 unique glycoPSMs were collectively identified in the DDA library and Byonic search, but not in the DIA-search. As this latter method is still an experimental (non-reviewed) workflow, we are not certain how reliable it is.

Minor Comments:

-Figures are not in the correct order. **We apologize for that and have now checked this in the revised manuscript**

-Figure 3 is incorrectly referenced in the text (“Figure 3D and 3E, Suppl...”)—it should be Figure 2.

-Figure 6A is incorrectly referenced—it should be Figure 5A.

-Optional minor Comment: The authors mention an increase in unique PSMs for non-glycopeptides in enriched plasma compared to crude plasma. This is an interesting finding. What are the additional proteins they identified?

We agree it is an interesting finding indeed, and we hypothesize that it is caused by a combination of specific enrichment of hydrophilic peptides, combined with non-specific enrichment of high abundant peptides, which effectively seems to reduce the dynamic range of the plasma sample. Looking at the non-glycosylated peptides only, we can match them to 319 proteins cumulatively identified in enriched plasma (using standard DIA method), compared to 251 proteins cumulatively identified in crude plasma (again, using standard DIA method). Of these, 10 were uniquely identified in normal plasma, and 78 were uniquely identified in enriched plasma, of these were 22 immunoglobulin light chain variable domains, the other proteins include: Factor 7, S100 calcium binding protein A4 (S100A4), Apolipoprotein C4 (APOC4), bone morphogenetic protein 2 (BMP2), and Interleukin 20 (IL20).

Reviewer #3 (Remarks to the Author):

Thank you for co-reviewing our work.

Reviewer #4 (Remarks to the Author):

In this manuscript, the authors utilized the latest Astral instrument with narrow-window DIA for plasma glycopeptide identification, which represents a new application of the Astral platform. However, I believe that this paper mainly presents one application of the Astral narrow-window DIA without addressing the critical challenges in glycopeptide DIA identification. It lacks systematic optimization and comprehensive comparison.

We thank the reviewer for their constructive comments and critical questions. We here address their major concerns regarding parameter optimization on the Orbitrap Astral, and cross validation of the data analysis using FragPipe.

1. One of the key advantages of DIA analysis is its ability to avoid the random selection of precursor ions in DDA. However, since DIA captures all MS2 data within a given window, a significant challenge lies in building the spectral library and ensuring precise matching and interpretation. While narrow-window DIA theoretically reduces chimeric spectra, the manuscript does not provide any validation of its accuracy, nor does it compare the results to existing glycopeptide DIA analyses or DDA results.

To address the reviewer's concern, we looked further into the occurrence of chimeric spectra, both in DDA using windows of 1.6 Th and 3 Th, ("Next, we aimed ... is still low" – page 6-7, and Supplemental Figure 1) and in larger isolation windows using DIA with 3, 6 and 12 Th ("Next, the effect ... 3 Th Isolation window" – page 7, and supplemental figure 2). Here we identified that the chimeric spectra increase along with the window size. As our approach mainly relies on DDA-based data analysis in a spectrum centric manner, our aim was to keep the number of chimeric spectra as low as possible, which was around 2.5% of annotated scans with 3 Th.

Furthermore, we added additional analysis using FragPipe both with DIA-Umpire and the experimental library-based DIA method, which is described on p 17-19 and the new Supplemental Figure S9. Both workflows are expected to yield less identifications, however the overlap we find is quite reasonable (92% with DIA-Umpire and 61% with library based DIA in FragPipe). A summary of the results can be found in the answer to the final question of reviewer 2.

2. Another key advantage of DIA is the ability to perform simultaneous quantification alongside identification. However, the authors did not present any quantitation results. Considering the identification level in this work does not seem to surpass current glycopeptide identification techniques significantly, the quantitative results and their accuracy should also be demonstrated.

We respectfully disagree with the reviewer that the identifications do not surpass current glycopeptide identification techniques when employed on human plasma. In plasma, the number of identifications will be lower than when using for example cell derived materials. This is because plasma has a high dynamic range, which typically limits the amount of unique PSMs that can be found. Other recent studies identified in plasma mostly less than 300 unique glycopeptides, which is more than 10 times less (Wessels et. al 2023 and Mao et al. 2022). However, since the experimental conditions are so different from the ones we used, including the instrument, enrichment method, and measurement method (DIA vs DDA), it remains rather difficult to make a direct comparison. For this reason, we have added in the revision, upon request of nearly all reviewers, additional data from DDA experiments on the Orbitrap Astral with the same sample and the same LC column and gradient. We specified this further in the discussion (“These 181 proteins ... to the DDA data” – page 17). In summary, we elaborate on the number of glycopeptides identified in the mentioned studies, and mention that the DIA method increases the identification of glycopeptides with 50% compared to the DDA method.

Furthermore, there is so far very limited software available to do quantification for DIA glycopeptide data. Thus, we did for the revision a pilot experiment, using a not yet peer-reviewed workflow in FragPipe and DIA-NN, which showed highly reproducible quantification over injection replicates (“An additional feature ... by peer review” – page 16, and Supplemental Figure 9D-E).

3. As a novel application for glycopeptide analysis, systematic optimization of the Astral instrument parameters is crucial. Although the authors mention some parameters, such as "a downside of shifting the MS1 range," I regret to note that beyond exploring different fragmentation energies (which is achievable on any Orbitrap instrument post-Fusion), the manuscript does not show results from other systematic parameter optimizations.

We respectfully disagree, as the following parameters were optimized: NCE, precursor m/z range, isolation window size and injection time. The latter 3 have large consequences for the cycle time, which we remained constant at 940 ms to achieve enough points per peak. One major concern we wanted to deal with was to minimize the number of chimeric spectra while having enough injection time to maximize spectral quality, which is why we decided on a 3 Th window and 4 ms scan time. To make this more clear, we have now included additional data on the window optimization into the results on page 6-7 (“In previously published...4 ms injection time”). and Supplemental Figure S2 in the revised manuscript. From this, it can be seen that increasing the window size to 6 and 12 Th will increase the number of chimeric spectra (from ~2.5% with 3 Th windows, to ~5.5% and ~9% for 6 and 12 Th windows, respectively). The Byonic score does not seem to increase or decrease, suggesting that spectral quality is not greatly affected. We do want to stress that library-free DIA is not yet well established for glycoproteomics and the availability of such algorithms might require different optimizations.

4. After applying the 950-1655 window, the authors mention a reduction in non-glycopeptide identifications. What about the number of glycopeptide identifications?

The number of glycopeptide identifications increased substantially compared to the ‘standard DIA precursor range’ of 380-980, as depicted in Figure 2A. Furthermore, compared to a ‘full window’ (m/z 300-2000), the number also increased by about 15%, which we now show using newly acquired DDA data in Supplemental Figure S1.

5. Why did the authors choose a library annotated for blood proteins instead of a comprehensive human protein library?

In Byonic we used a focused database containing blood (=serum) proteins to limit the search space and decrease search time, as is done frequently by other groups as well (for example: Riley et al, Nat Comm 2019). Another reason is that proteins in plasma can be different from the in-cell variants of the same gene, for example IgGs. The canonical IgG sequence in UniProt is the membrane bound sequence, which does not harbor the N-glycosylation site, but adding all isoforms of all proteins would increase the search space by too much. This dedicated plasma database does contain high abundant isoforms of plasma proteins, as deposited in UniProt or NexProt, with allele frequencies above 5% in the population.

6. Page 8: “Almost all identified glycosylated PSMs found...”—What does "almost all" mean? 80%? 90%? This should be quantified.

We clarified this in the following way: “Almost all identified glycosylated PSMs found in the standard DIA method (~90%) were detected in at least one of the nGlycoDIA methods as well.”

7. Page 8: “..., indicating high method robustness (Figure 3D and 3E, Supplemental Figure S4A and S4B)” should refer to Figures 2D and 2E.

Thank you for the comment, we checked all the figure referencing and corrected them.

8. In “Setting up narrow-window nGlycoDIA,” it states 955-1650, while Table 1 and the Supplementary Figure S1 show 950-1655. These discrepancies should be corrected.

This has now been corrected.

9. The background does not sufficiently summarize the current state of glycopeptide DIA analysis. Key works such as GproDIA and GlycoDIA are missing and should be cited.

We thank the reviewer for pointing us to these publications and added them to our manuscript accordingly:

“For O-glycoproteomics, glyco-DIA has been developed as platform that allows library generation based on DDA data, with subsequent analysis and quantification of DIA data²³. For N-glycoproteomics, GproDIA has been developed, which relies on pGlyco for library generation for subsequent DIA analysis and quantification²⁴” – **page 2. And “An additional advantage of DIA data is the ability to perform quantification based on the MS². This has previously been exploited by Yang *et al.* in their GproDIA algorithm²⁴.” – page 19.**

Reviewer #4 (Remarks on code availability):I did not run the entire dataset to check for reproducibility and only tested a small subset of the data. The code includes a README file with sufficient instructions for installation and running.

REVIEWERS' COMMENTS

Reviewer #1 (Remarks to the Author):

have carefully reviewed your rebuttal and the corresponding revisions made to the manuscript. I appreciate the effort you have put into addressing all the comments and concerns raised during the review process. Your thorough and well-structured responses have significantly improved the clarity and rigor of the manuscript.

The modifications you have implemented enhance the overall quality of the study, ensuring that your findings are well-supported and effectively communicated. I am satisfied with the revisions and have no further concerns. The manuscript is now suitable for publication in its current form.

Thank you for your diligence in refining the work. I look forward to seeing your study published.

We thank the reviewer for their positive response.

Reviewer #2 (Remarks to the Author):

The authors have addressed the reviewers' comments and improved the manuscript. I recommend acceptance of this work but have one minor comment that arose from the revision, which could be incorporated into the discussion: it is interesting to note that the identifications increased with window size. Are these results reliable, and does this suggest that this approach could also be adapted to other (older) MS platforms that typically require wider windows? Including a discussion on this point could be valuable, as it would emphasize that this approach is not limited to users of the Astral platform.

We thank the reviewer for the positive response. We think that using larger windows is conceivable and could be translated to other instruments, as has by Pradita et al. (JPR, 2024). We describe this in the discussion:

“However, as we demonstrated, the number of chimeric spectra increased substantially by extending the window size, while the overall Byonic scores remained similar. As our workflow relied primarily on the use of Byonic, we therefore decided to minimize the occurrence of chimeric spectra. **However, even if chimeric spectra increased, the data analysis of wider windows was feasible, enabling the translation of this method to other instruments that require wider isolation windows. This has been demonstrated recently**, when a similar Byonic strategy was used on DIA data with 16 Th windows³⁰. However, this was performed on a purified protein, thus the data was much less complex, and the chance of chimeric spectra is lower than when a complex sample such as plasma is used.”

Reviewer #3 (Remarks to the Author):

I co-reviewed this manuscript with one of the reviewers who provided the listed reports. This is part of the Nature Communications initiative to facilitate training in peer review and to provide

appropriate recognition for Early Career Researchers who co-review manuscripts.

We thank the reviewer for co-reviewing our manuscript and support.

Reviewer #4 (Remarks to the Author):

The authors have addressed most of my concerns. The revisions significantly strengthen the manuscript, and the improvements in clarity and data presentation are highly appreciated. The additional analysis of chimeric spectra and comparisons across different isolation windows, coupled with cross-validation using FragPipe, effectively address the concern regarding the accuracy of the DIA method. The authors' efforts to minimize chimeric spectra by optimizing window size and injection time are commendable. The clear presentation of results, such as achieving only 2.5% chimeric spectra at a 3 Th window, inspires confidence in the methodology. I have no further concerns regarding this point.

We thank the reviewer for their positive comments and the time spent reviewing our manuscript.

The authors have also provided a reasonable explanation for the lack of direct quantification results, acknowledging the current limitations of available software for DIA glycopeptide quantification. Furthermore, the rationale for using a blood protein-specific library to reduce search space and improve efficiency is acceptable. I believe the limitation reflects the broader challenges in developing tools tailored for DIA glycopeptide analysis, which applies not only to this study but to the field as a whole. The authors should discuss this issue in the manuscript.

We fully agree that software is still a bottleneck. Byonic performs in our hands well, but is time-inefficient, making us adopt a smaller search space lacking some possible isoforms found in blood not present in the canonical human proteome database. Including all isoforms would lead to days of search time with Byonic. When using Fragpipe we were able to use the entire human database, including isoforms, and it was fast and efficient, but did not give that many IDs yet.

Regarding the identification numbers, it is undeniable that this work does not represent the forefront in terms of quantity, as many glycoproteomics studies report higher identification counts. While I do think the focus of glycoproteomics is shifting away from merely pursuing higher identification numbers, the authors should discuss this point in the manuscript, as the primary goal of DIA analysis is to enable deeper and broader insights into glycopeptide characterization.

We respectfully disagree with the reviewer, as we were not able to find such studies targeting human plasma, specifically glycopeptide enriched human plasma. The recent studies we found that use enriched plasma all identified a 10-fold lower amount of glycopeptides, using 2-3 times longer measuring times. These are all mentioned in the discussion. Additionally, higher identification numbers, especially in the case of plasma glycoproteomics, are bound to enrichment and fractionation protocols, and we demonstrate in this paper a 2-fold increase compared to DDA using the same sample on

the same instrument with the same LC-conditions. Maybe the reviewer refers to non-plasma samples?

Reviewer #4 (Remarks on code availability):

I did not reviewed the code in this revision.

It is available